# VCR: Variance-aware Channel Recalibration Network for Low Light Image with Distribution Alignment

## Abstract

Most sRGB-based LLIE methods suffer from entangled luminance and color, while the HSV color space offers insufficient decoupling at the cost of introducing significant red and black noise artifacts. Recently, the HVI color space has been proposed to address these limitations by enhancing color fidelity through chrominance polarization and intensity compression. However, existing methods could suffer from channel-level inconsistency between luminance and chrominance, and misaligned color distribution may lead to unnatural enhancement results. To address these challenges, we propose the Variance-aware Channel Recalibration Network for Low Light Image with Distribution Alignment (VCR), a novel framework for low-light image enhancement. VCR consists of two main components, including the Channel Adaptive Adjustment (CAA) module, which employs variance-guided feature filtering to enhance the model's focus on regions with high intensity and color distribution. And the Color Distribution Alignment (CDA) module, which enforces distribution alignment in the color feature space. These designs enhance perceptual quality under low-light conditions. Experimental results on several benchmark datasets demonstrate that the proposed method achieves state-of-the-art performance compared with existing methods.

## 1 Introduction

Low-Light Image Enhancement (LLIE) (Hu et al., 2025; Zhou et al., 2023; Kim et al., 2022; Zeng et al., 2025; Wang et al., 2025; Jiang et al., 2024) aims to improve the brightness, contrast, and visibility of details in images captured under poor illumination. Due to the inherent limitations of imaging sensors in low-light conditions, such images often suffer from severe noise and degradation. As a fundamental preprocessing step, low-light enhancement can benefit a wide range of downstream vision tasks such as object detection (Zou et al., 2023), instance segmentation (Bolya et al., 2019), and image matching (Cheng et al., 2025). However, standard RGB-based (sRGB) methods often struggle due to the strong coupling between color and luminance, leading to unnatural brightness or color distortions after enhancement. Effectively decoupling color and luminance remains an open and challenging problem.

To tackle the above challenges, low-light image enhancement methods can be grouped into three categories. Traditional approaches (Wang et al., 2022) in the sRGB space often suffer from strong coupling between luminance and color, resulting in unnatural brightness and severe color distortions. Inspired by the Kubelka-Munk (Gevers et al., 2012) theory, some methods (Guo & Hu, 2023) adopt the HSV color space to decouple brightness from chrominance, enabling more controllable enhancement. However, HSV-based methods tend to introduce artifacts such as red discontinuity and black plane noise, which degrade image quality. To address these issues, CIDNet (Yan et al., 2025) proposes the HVI color space for low-light conditions, which improves both color fidelity and visual naturalness. Specifically, it mitigates red-channel artifacts through hue-saturation plane polarization and suppresses black noise via a learnable intensity compression transform. However, due to the varying dynamic ranges of luminance and chrominance, different feature channels may focus unevenly across the space, leading to channel-level feature misalignment and reduced enhancement accuracy.

Figure 1: (a) Different feature channels focus on different regions. By selectively filtering channels, regions with consistent brightness and color distributions are enhanced, leading to improved overall enhancement performance. (b) Consistency in color space distribution helps the image achieve a more natural appearance, leading to more realistic and visually pleasing enhancement results.

Building on the discussion of HVI-based methods, we identify two key challenges that remain unsolved for achieving more accurate and robust low-light image enhancement: **(1)** *How to selectively filter luminance and chrominance channels to focus on regions with strong intensity and color variation.* Although previous methods (Yan et al., 2025) mitigate red discontinuity and black plane noise via chrominance polarization and learnable intensity compression, inconsistencies may still arise due to the spatial misalignment of luminance and color focus across different channels. This might lead to artifacts or color shifts during enhancement. As illustrated in Figure 1(a), a more effective channel-wise feature filtering mechanism could suppress noisy or irrelevant activations in dark regions and improve enhancement quality. **(2)** *How to optimize the color distribution of enhanced images.* We observe that color distortion is fundamentally linked to the distribution of chrominance in the feature space. While earlier methods decouple luminance and color, they often overlook the structure of color distribution itself. As shown in Figure 1(b), imposing a consistency constraint on color distribution could play a crucial role in producing clearer and more natural subjective results under low-light conditions.

To effectively address the challenges of channel-level inconsistency and color distortion in low-light image enhancement, we propose Variance-aware Channel Recalibration Network with Distribution Alignment (VCR). Our approach introduces two key modules: the Channel Adaptive Adjustment (CAA) module and the Color Distribution Alignment (CDA) module. In CAA, we first design a Variance-aware Channel Filtering (VCF) stage to identify and mask channels with large variance between luminance and chrominance distributions. These filtered features are then adaptively fused with the original ones, allowing the model to focus on regions of joint distributional saliency of luminance and color while preserving feature independence to suppress artifacts. Subsequently, we introduce the Triplet Channel Enhancement (TCE) stage (Zhou et al., 2021; Hou et al., 2021), which builds inter-channel and spatial dependencies through rotation-based operations followed by residual transformations, enabling more robust channel-wise feature enhancement. In CDA, we enforce a distribution consistency constraint between the enhanced image and a real-scene reference in the color feature space. The above alignment allows the model to learn a more realistic color distribution and effectively reduces color shifts in low-light conditions.

Our contributions can be summarized as follows:

- We propose a novel framework, Variance-aware Channel Recalibration Network for Low-Light Image with Distribution Alignment (VCR), which achieves superior performance on the low-light image enhancement task.
- We design the Channel Adaptive Adjustment (CAA) module to adaptively filter and enhance luminance and chrominance features at the channel level, improving perceptually realistic lighting and color characteristics. Additionally, we introduce the Color Distribution Alignment (CDA) module, which enforces consistency in the color feature distribution, leading to clearer and more natural results.
- Extensive experiments and ablations on ten benchmark datasets demonstrate the effectiveness and generalization ability of our method, establishing a new state-of-the-art in low-light image enhancement.

## 2 RELATED WORKS

In this section, we review existing approaches for low-light image enhancement.

**Traditional Methods.** Early low-light enhancement relied on heuristic image processing techniques that do not require training data. Histogram equalization (Pizer et al., 1987) and gamma correction (Rahman et al., 2016) amplify contrast and brightness by redistributing pixel intensities, but they disregard scene illumination and may produce over-enhanced or washed-out results. Retinex-based approaches (Land & McCann, 1971; Rahman et al., 2004) decompose an image into illumination and reflectance components, refining the illumination estimate with structure prior. While more physically grounded, these methods assume ideal inputs and suffer from noise amplification and color distortion in real-world low-light scenarios.

**Learning-Based Methods.** The advent of deep learning transformed low-light enhancement into a data-driven task. Supervised models such as RetinexNet (Wei et al., 2018) and KinD (Zhang et al., 2019) integrate Retinex decomposition within a CNN, but their reliance on accurate illumination estimation can amplify noise and lead to color shifts. ZeroDCE (Guo et al., 2020) and RUAS (Liu et al., 2021) bypass explicit decomposition by learning pixel-adaptive curves or spatial structure search, which may introduce artifacts and unstable chrominance. Flow-based LLFlow (Wang et al., 2022) achieves high restoration fidelity through normalizing flows but incurs substantial computational cost and depends on paired supervision. GAN-based methods such as EnlightenGAN (Jiang et al., 2021) enhance perceptual realism via adversarial training but may produce unrealistic textures. SNR-Aware networks (Xu et al., 2022) incorporate noise priors to suppress artifacts, which may suffer from the color unconsistency. Recent transformer-based architectures, including LL-Former (Wang et al., 2023) and RetinexFormer (Cai et al., 2023), capture long-range dependencies but do not explicitly enforce channel-level alignment. Bread (Guo & Hu, 2023) decouples the entanglement of noise and color distortion by using YCbCr color space, while GSAD (Hou et al., 2023) builds a global structure-aware diffusion process to improve the quality, which may exhibit issues such as local overexposure or color shift. QuadPrior (Wang et al., 2024) introduces four physical priors establishing constraints for enhancing illumination; however, the complex optimization process results in excessive parameter count and computational cost. CIDNet (Yan et al., 2025) introduces HVI color-space to mitigate red discontinuities and black noise. Despite the considerable advancements made by existing methods, they typically neglect the distributional disparities of inter-channels, which may result in uneven enhancement and residual color biases.

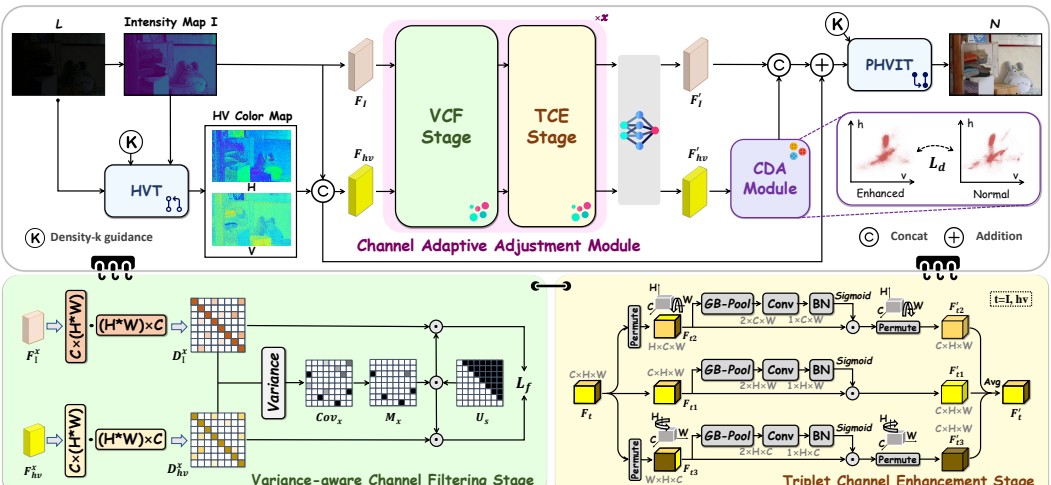

Figure 2: The overall pipeline of the VCR process begins by transforming the input into the HVI space. Next, it is processed by the Channel Adaptive Adjustment module, which includes Variance-aware Channel Filtering and Triplet Channel Enhancement stage. These techniques aim to emphasize regions with a high consistency of luminance and chromaticity by filtering and enhancing the channels. After this recalibration, the features are refined, and the HV components are aligned with ground-truth statistics via the Color Distribution Alignment (CDA) module to mitigate color shifts. Finally, the enhanced output is reconstructed in the sRGB color space.

## 3 METHODS

The proposed variance-aware channel recalibration scheme with distribution alignment is illustrated in Figure 2. The input image is mapped into the HVI color space to disentangle luminance from chromaticity and fed into the Channel Adaptive Adjustment module, which comprise a Variance-aware Channel Filtering stage to suppress discrepancies in feature distributions and a Triplet Channel Enhancement stage to build inter-channel and spatial dependencies. Upon recalibration, both features pass through the enhancement network, where the HV channels undergo distribution alignment with the ground truth via the Color Distribution Alignment (CDA) module subsequently. Finally, the enhanced HVI representation is mapped back to the RGB domain. In this section, we explain the role of the HVI transformation and each submodule.

### 3.1 HVI COLOR SPACE

In the standard sRGB color space, image brightness and chromatic information are tightly coupled across the three color channels, which may disrupt the perceived illumination or color balance of the entire image when making adjustments to any individual channel. Although the HSV color space separates intensity from chromaticity, it inadvertently amplifies noise in regions of extreme red and near-black areas, producing pronounced "red-discontinuity" and "black-plane" artifacts during enhancement. To address the above limitations, the HVI color space has been proposed to alleviate

---

**Algorithm 1:** HVI Color Transform and Perceptual-inverse HVI Transform

---

**Input:** sRGB image $I$ with channels $I_R, I_G, I_B$
**Output:** Enhanced sRGB image $\tilde{I}$
`;// k: density-k;` $\alpha_S, \alpha_I$: `scaling;` $\varepsilon = 10^{-8}$
**foreach** *pixel $x$ in $I$* **do**
    $I_{\max}(x) \leftarrow \max\{I_R(x), I_G(x), I_B(x)\},;$
    $I_{\min}(x) \leftarrow \min\{I_R(x), I_G(x), I_B(x)\};$
    **if** $I_{\max}(x) = 0$ **then**
        $S(x) \leftarrow 0, H(x) \leftarrow 0$
    **else**
        $S(x) \leftarrow \dfrac{I_{\max}(x) - I_{\min}(x)}{I_{\max}(x)};$
        **if** $I_{\max}(x) = I_R(x)$ **then**
            $H(x) \leftarrow \left( \dfrac{I_G(x) - I_B(x)}{I_{\max}(x) - I_{\min}(x)} \right) \bmod 6;$
        **else if** $I_{\max}(x) = I_G(x)$ **then**
            $H(x) \leftarrow 2 + \dfrac{I_B(x) - I_R(x)}{I_{\max}(x) - I_{\min}(x)};$
        **else**
            $H(x) \leftarrow 4 + \dfrac{I_R(x) - I_G(x)}{I_{\max}(x) - I_{\min}(x)};$
    $C_k(x) \leftarrow k\sqrt{\sin\left(\frac{\pi I_{\max}(x)}{2}\right) + \varepsilon}; \hat{H}(x) \leftarrow C_k(x)S(x)\cos\left(\frac{\pi H(x)}{3}\right);$
    $\hat{V}(x) \leftarrow C_k(x)S(x)\sin\left(\frac{\pi H(x)}{3}\right);$
Obtain HVI representation $(I_{\max}, \hat{H}, \hat{V})$;
Apply Channel Adaptive Adjustment (Variance-aware filtering + Triplet Channel Enhancement) and
  Color Distribution Alignment (CDA) on $(I_{\max}, \hat{H}, \hat{V})$ to obtain $(I'_{\max}, \hat{H}', \hat{V}')$;
**foreach** *pixel $x$* **do**
    $\hat{h}(x) \leftarrow \dfrac{\hat{H}'(x)}{C_k(x) + \varepsilon}; \hat{v}(x) \leftarrow \dfrac{\hat{V}'(x)}{C_k(x) + \varepsilon};$
    $H(x) \leftarrow \dfrac{1}{2\pi} \arctan\left(\hat{v}(x)/\hat{h}(x)\right) \bmod 1;$
    $S(x) \leftarrow \alpha_S\sqrt{\hat{h}^2(x) + \hat{v}^2(x)}; V(x) \leftarrow \alpha_I I'_{\max}(x);$
    $(\tilde{I}_R(x), \tilde{I}_G(x), \tilde{I}_B(x)) \leftarrow \text{HSV2sRGB}(H(x), S(x), V(x));$
**return** $\tilde{I}$;

---

inherent color noise, which is composed of three channels: $I_{\max}$, $\hat{H}$, and $\hat{V}$, designed to mitigate the artifacts introduced by the HSV representation. Here, $C_k(x)$ denotes a learnable intensity collapse function that remaps the maximum intensity $I_{\max}(x)$ for stabilizing low-light responses. The parameter $k$, termed *density-k*, controls the density of black-plane points in HVT/PHVIT, thereby balancing noise suppression and detail preservation (shown in Algorithm 1 and detailed in Appendix A.2).

## 3.2 Channel Adaptive Adjustment Module

Since different feature channels attend to different regions, we aim to focus on areas with high consistency in intensity and color distribution. Thus, we design a Channel Adaptive Adjustment (CAA) module that filters and enhances channel-wise attention to regions with high luminance and chrominance distributions, thereby recalibrating feature representations and improving inter-channel distributional consistency. Specifically, CAA sequentially applies two modules at the channel level: Variance-aware Channel Filtering (VCF) for selective suppression, followed by Triplet Channel Enhancement (TCE) for targeted amplification.

### 3.2.1 Variance-aware Channel Filtering Stage.

As introduced in the previous section, after converting an sRGB image into the HVI space (through HVT), the intensity map is obtained using Equation (1), and the HV chromaticity maps are derived via Equation (4). By applying a concatenation operation, we obtain the intensity feature $F_I \in \mathbb{R}^{H \times W \times C}$ and the chromaticity feature $F_{hv} \in \mathbb{R}^{H \times W \times C}$.

Most low-light enhancement networks adopt batch normalization (BN) by default, which relies on training set statistics and is sensitive to distribution shifts. To enhance consistency in intensity and color distribution, we replace BN with instance normalization (IN), which normalizes each sample independently by subtracting its own mean and dividing by its standard deviation. We further consider the information embedded in feature covariance, which is not addressed by instance normalization (IN). Our goal is to suppress covariance components that are sensitive to intensity and color variation through channel-wise modulation, thereby focusing on regions with high-value intensity and color distributions.

To this end, we compute the covariance matrices of intensity and chromaticity features, denoted as $D_I^x \in \mathbb{R}^{C \times C}$ and $D_{hv}^x \in \mathbb{R}^{C \times C}$, respectively. These covariance matrices are calculated as follows:

$$\mathbf{D}_I^x = \frac{1}{HW} \left( \mathbf{F}_I^{x\top} \cdot \mathbf{F}_I^x \right), \quad \mathbf{D}_{hv}^x = \frac{1}{HW} \left( \mathbf{F}_{hv}^{x\top} \cdot \mathbf{F}_{hv}^x \right),$$ (1)

where $x$ denotes the iteration index of the CAA module. We then compute the cross-covariance matrix $Cov_x$ between $D_I^x$ and $D_{hv}^x$:

$$\mu_x = \frac{1}{2}(D_I^x + D_{hv}^x),$$ (2)

$$Cov_x = \frac{1}{2}((D_I^x - \mu_x)^2 + (D_{hv}^x - \mu_x)^2).$$ (3)

The element $Cov_x(i,j)$ in the covariance matrix measures how sensitive the $i$-th and $j$-th channels are across intensity and chromaticity. More specifically, a higher $Cov_x(i,j)$ indicates that the feature correspondence between the $i$-th intensity channel and the $j$-th chromaticity channel is more likely to exhibit low distributional consistency, and vice versa. Feature channels with significantly large variance values may tend to focus on regions that are not characterized by strong intensity or chromaticity distributions, which is unfavorable for the low-light image enhancement task. These components should be suppressed through constraints; therefore, we design $L_{\text{VCF}}$ to filter out parts with excessively high variance values:

$$\mathcal{L}_{\text{VCF}} = \frac{1}{X} \sum_{x=1}^{X} \left( \|D_I^x \odot M_x\|_1 + \|D_{hv}^x \odot M_x\|_1 \right),$$ (4)

where x represents the layer processed by CAA. The covariance matrix $Cov_x$ is divided into three groups based on the width of variance. We select the one with the highest variance value for masking. Then, we multiply the selected $M_x$ by a strict upper triangular matrix $\mathbf{U}_s$, update $M_x$, and then apply it to $\mathbf{D}_I^x$ and $\mathbf{D}_{hv}^x$. Since the covariance matrix is symmetric, as training progresses, it may cause the network to overfit the statistical characteristics of a specific modality, making it difficult to generalize to other low-light scenarios. By optimizing only the upper triangular part, we can prevent the model from overly relying on the statistical information of a particular modality, reduce redundant information and enhance feature independence. Furthermore, to ensure feature completeness and prevent important information from being masked out, we also integrate the filtered features with the original features, obtaining the updated intensity feature $F_I \in \mathbb{R}^{H \times W \times C}$ and chromaticity feature $F_{hv} \in \mathbb{R}^{H \times W \times C}$.

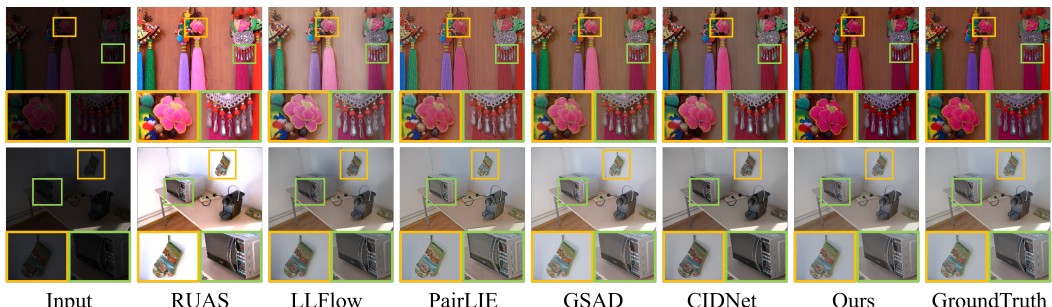

| Input | RUAS | LLFlow | PairLIE | GSAD | CIDNet | Ours | GroundTruth |

Figure 3: Visual results of various methods on the LOL dataset. Regions highlighted by green and yellow boxes indicate differences in local details.

### 3.2.2 TRIPLET CHANNEL ENHANCEMENT STAGE.

Subsequently, the Triplet Channel Enhancement (TCE) stage refines the recalibrated features by capturing complementary channel and spatial correlations. We first adjust the feature channel order to $F_t \in \mathbb{R}^{C \times H \times W}$, where $t \in \{I, hv\}$. Three parallel branches are designed for the intensity–chromaticity features: two capture cross-dimensional interactions between the channel dimension $C$ and the spatial dimensions $H$ or $W$, and the third establishes spatial attention. The final output is obtained by averaging the outputs of all three branches. This design optimizes attention computation by effectively capturing the relationships between spatial and channel dimensions.

In TCE, we model the dependencies between the dimensions $(H, W)$, $(C, H)$, and $(C, W)$ to better exploit the input features, thereby enhancing feature representation, improving consistency in intensity–chromaticity distribution, and increasing robustness in the low-light image enhancement task. Given an input feature $F_t \in R^{C \times H \times W}$, three rotated views are constructed:

$$
\begin{cases}
F_{t1} = Permute(F_t; H, C, W), \\
F_{t2} = Permute(F_t; W, C, H), \\
F_{t3} = F_t.
\end{cases}
\tag{5}
$$

The Global-Best Pooling layer (GB-Pool) combines max pooling (Murray & Perronnin, 2014) and average pooling (Sun et al., 2017) to reduce the first dimension of the tensor to two, preserving feature richness while improving computational efficiency. Specifically, the GB-Pool operation is defined as:

$$
GB\text{-}Pool(F_{ty}) = [\, MaxPool_{0d}(F_{ty}),\ AvgPool_{0d}(F_{ty}) \,],
\tag{6}
$$

where $F_{ty}$ represents the image features of the $t$-th feature channel and $y$-th branch, and $0d$ denotes pooling along the first dimension. $y \in \{1, 2, 3\}$ represents the different branches.

Taking $F_{t2}$ as an example, we establish an interaction between the height and channel dimensions. We rotate $F_t$ counterclockwise by $90°$ along the $W$-axis. The rotated tensor $F_{t2}$ is passed through the GB-Pool layer, reducing it to shape $2 \times C \times W$. A convolution layer with kernel size $k \times k$, followed by a batch normalization (BN) layer, normalizes the output to shape $1 \times C \times W$. The tensor is then passed through a sigmoid activation layer $\sigma$ to obtain attention weights, which are applied to $F_{t2}$. Finally, the output is rotated $90°$ clockwise along the $W$-axis to match the original shape of $F_t$. Similarly, $F_{t3}$ is obtained by rotating $F_t$ counterclockwise along the $H$-axis, and $F_{t1}$ is not rotated.

This process is expressed as:

$$
F_t' = \frac{1}{3} \sum_{y=1}^{3} \left( F_{ty} \cdot \sigma \big( \mathrm{Conv}(GB\text{-}Pool(F_{ty})) \big) \right),
\tag{7}
$$

where $\sigma$ is the sigmoid function, and Conv is a 2D convolution layer with kernel size $k$. To preserve feature independence, the original features are fused with the enhanced ones via residual connections.

### 3.3 COLOR DISTRIBUTION ALIGNMENT MODULE

Enhancing images often leads to color shifts, which are closely tied to the distribution of chromatic features. To address this issue, we decouple intensity and color through a carefully designed pipeline and impose explicit constraints on the chrominance distribution, significantly improving color fidelity and mitigating color shift artifacts.

To further reduce residual color distortion and enforce realistic chrominance statistics, we introduce a Color Distribution Alignment (CDA) module that aligns the enhanced HV features with ground truth references in a distributional sense. After the CAA stage, a dual-branch network separately enhances luminance and denoises

Figure 4: Qualitative comparison of enhancement results on the unpaired dataset in difficult condition, generated by various methods.

chrominance, and uses cross-attention to fuse complementary information, leading to improved low-light image enhancement.

After this sequence of operations, we denote the enhanced HV feature map and the corresponding ground truth as $F'_{hv} \in \mathbb{R}^{2C \times H \times W}$ and $F^{gt}_{hv} \in \mathbb{R}^{2C \times H \times W}$, respectively. We compute channel-wise probability distributions using a temperature-scaled softmax function:

$$\begin{cases} p_{c,a} = \dfrac{\exp\left(F'_{hv}(c,a)/\tau\right)}{\sum_{b=1}^{N} \exp\left(F'_{hv}(c,b)/\tau\right)}, \\ q_{c,a} = \dfrac{\exp\left(F^{gt}_{hv}(c,a)/\tau\right)}{\sum_{b=1}^{N} \exp\left(F^{gt}_{hv}(c,b)/\tau\right)}, \end{cases} \tag{8}$$

where $c = 1, \ldots, 2C$ is the channel index, $a = 1, \ldots, N$ is the spatial location index, and $N = H \cdot W$. The Color Distribution Alignment (CDA) loss is defined as the sum of Kullback–Leibler (Johnson et al., 2001) divergences across all channels:

$$\mathcal{L}_{\text{CDA}} = \sum_{c=1}^{2C} \sum_{a=1}^{N} p_{c,a} \log \frac{p_{c,a}}{q_{c,a}}. \tag{9}$$

Minimizing $\mathcal{L}_{\text{CDA}}$ encourages the model to produce chrominance features that closely follow the distribution of well-exposed reference images. By enforcing alignment in the probability space rather than raw values, CDA captures subtle color statistics and structure more effectively. This helps reduce chrominance drift and improves the visual realism and perceptual quality of enhanced images, particularly in challenging low-light conditions.

Finally, to reconstruct the final image, the HVI representation is converted back to the HSV color space via the Perceptual-inverse HVI Transformation (PHVIT), as formulated in Equations (6) and (7). The resulting HSV image is then transformed into the sRGB domain to obtain the enhanced output.

## 3.4 LOSS FUNCTION

To constrain the training of the proposed framework, we employ a comprehensive loss that integrates the primary reconstruction loss in both RGB and HVI spaces with the VCF loss and CDA loss. Concretely, let $I_{\text{out}}$ and $I_{\text{gt}}$ denote the enhanced and ground-truth images in the RGB domain, and let $I_{\text{out}}^{\text{HVI}}$ and $I_{\text{gt}}^{\text{HVI}}$ denote their counterparts in the HVI color space. We define the reconstruction loss as:

Table 1: Quantitative results of PSNR↑/SSIM↑ and LPIPS↓ on the LOL (v1 and v2) datasets. The best performance is in **red**, and the second best is in **blue**.

| Methods | Complexity | | LOLv1 | | | LOLv2-Real | | | LOLv2-Synthetic | | |
|---|---|---|---|---|---|---|---|---|---|---|---|
| | Params/M | FLOPs/G | PSNR↑ | SSIM↑ | LPIPS↓ | PSNR↑ | SSIM↑ | LPIPS↓ | PSNR↑ | SSIM↑ | LPIPS↓ |
| RetinexNet (Wei et al., 2018) | 0.84 | 584.47 | 18.915 | 0.427 | 0.470 | 16.097 | 0.401 | 0.543 | 17.137 | 0.762 | 0.255 |
| KinD (Zhang et al., 2019) | 8.02 | 34.99 | 23.018 | 0.843 | 0.156 | 17.544 | 0.669 | 0.375 | 18.320 | 0.796 | 0.252 |
| ZeroDCE (Guo et al., 2020) | 0.075 | 4.83 | 21.880 | 0.640 | 0.335 | 16.059 | 0.580 | 0.313 | 17.712 | 0.815 | 0.169 |
| RUAS (Liu et al., 2021) | 0.003 | 0.83 | 18.654 | 0.518 | 0.270 | 15.326 | 0.488 | 0.176 | 13.765 | 0.638 | 0.305 |
| LLFlow (Wang et al., 2022) | 17.42 | 358.4 | 24.998 | 0.871 | 0.117 | 17.433 | 0.831 | 0.315 | 24.870 | 0.919 | 0.067 |
| EnlightenGAN (Jiang et al., 2021) | 114.35 | 61.01 | 20.003 | 0.691 | 0.317 | 18.230 | 0.617 | 0.309 | 16.570 | 0.734 | 0.220 |
| SNR-Aware (Xu et al., 2022) | 4.01 | 26.35 | 26.716 | 0.851 | 0.152 | 21.480 | 0.849 | 0.163 | 24.140 | 0.928 | 0.056 |
| Bread (Guo & Hu, 2023) | 2.02 | 19.85 | 25.299 | 0.847 | 0.155 | 20.830 | 0.847 | 0.174 | 17.630 | 0.919 | 0.091 |
| PairLIE (Fu et al., 2023) | 0.33 | 20.81 | 23.526 | 0.755 | 0.248 | 19.855 | 0.778 | 0.317 | 19.074 | 0.794 | 0.230 |
| LLFormer (Wang et al., 2023) | 24.55 | 22.52 | 25.758 | 0.823 | 0.167 | 20.056 | 0.792 | 0.211 | 24.038 | 0.909 | 0.066 |
| RetinexFormer (Cai et al., 2023) | 1.53 | 15.85 | 27.140 | 0.850 | 0.129 | 22.794 | 0.840 | 0.171 | 25.670 | 0.930 | 0.059 |
| GSAD (Hou et al., 2023) | 17.36 | 442.02 | 27.605 | 0.876 | 0.092 | 20.153 | 0.846 | 0.113 | 24.472 | 0.929 | 0.051 |
| QuadPrior (Wang et al., 2024) | 1252.75 | 1103.20 | 22.849 | 0.800 | 0.201 | 20.592 | 0.811 | 0.202 | 16.108 | 0.758 | 0.114 |
| CIDNet (Yan et al., 2025) | 1.88 | 7.57 | **28.201** | **0.889** | **0.079** | **24.111** | **0.871** | **0.108** | **25.705** | **0.942** | **0.045** |
| Ours | 1.96 | 8.32 | **28.972** | **0.891** | **0.083** | 24.758 | **0.893** | **0.105** | **26.273** | **0.944** | **0.042** |

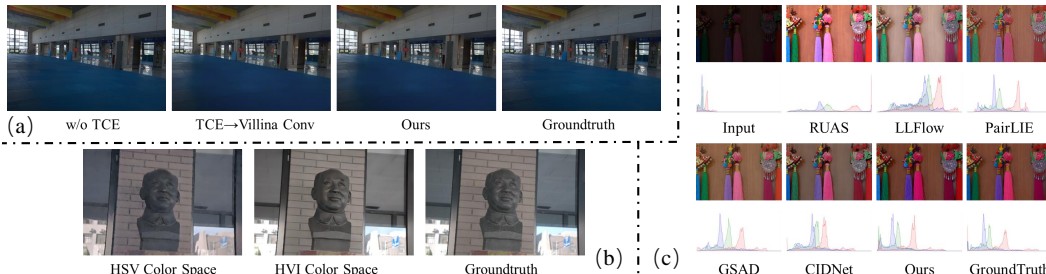

Figure 5: (a) Qualitative comparison of TCE module. (b) Ablation visualization results of different space. (c)Visual comparison between prior methods and ours. The top row presents the RGB images, while the bottom row shows the corresponding discrete color histograms.

$$\mathcal{L}_{\text{rec}} = \|I_{\text{out}} - I_{\text{gt}}\|_1 + \lambda_{\text{HVI}} \left\| I_{\text{out}}^{\text{HVI}} - I_{\text{gt}}^{\text{HVI}} \right\|_1, \tag{10}$$

where $\|\cdot\|_1$ denotes the $\ell_1$ norm and $\lambda_{\text{HVI}}$ is a weighting coefficient, set to 1 in our experiments. The overall loss function is then formulated as:

$$\mathcal{L}_{\text{total}} = \mathcal{L}_{\text{rec}} + \lambda_{\text{VCF}} \mathcal{L}_{\text{VCF}} + \lambda_{\text{CDA}} \mathcal{L}_{\text{CDA}}, \tag{11}$$

where $\lambda_{\text{VCF}}$ and $\lambda_{\text{CDA}}$ balance the contributions of the $\lambda_{\text{VCF}}$ and $\lambda_{\text{CDA}}$, respectively. It is worth noting that, during the initial training phase, we optimize exclusively with the reconstruction loss by setting $\lambda_{\text{VCF}} = 0$ and $\lambda_{\text{CDA}} = 0$. The weights of VCF and CDA losses set as $\lambda_{\text{VCF}} = 0.5$ and $\lambda_{\text{CDA}} = 0.5$, respectively.

## 4 EXPERIMENTS

### 4.1 DATASETS AND SETTINGS

**Datasets.** To validate the effectiveness of the proposed method, we conduct experiments on ten LLIE benchmark datasets, including five paired datasets: LOLv1 (Wei et al., 2018), LOLv2 (Yang et al., 2021)(including two subsets), SICE (Cai et al., 2018), and SID (Chen et al., 2018), and five unpaired datasets, including DICM (Lv et al., 2018), LIME (Guo et al., 2016), MEF (Ma et al., 2015), NPE (Wang et al., 2013), and VV (Vonikakis et al., 2018). LOLv1 dataset contains 485 paired images for training and 15 for testing. LOLv2 comprises two subsets: LOLv2-Real and LOLv2-Synthetic, containing 689 and 900 paired training images respectively, with each subset having 100 testing images. The SICE dataset contains 589 pairs of low-light and well-exposed images. In our experiments, we randomly selected 100 image pairs for testing, while the remaining 489 pairs were used for training and validation.

**Experiment Settings.** We implement the proposed method using PyTorch and train all models on a single NVIDIA RTX 3090 GPU. The optimizer is Adam (Kingma & Ba, 2014) with parameters $\beta_1 = 0.9$ and $\beta_2 = 0.999$. The initial learning rate is set to $1 \times 10^{-4}$ and is gradually reduced to $1 \times 10^{-7}$ using a cosine annealing schedule (Loshchilov & Hutter, 2016). During training, the batch size is consistently set to 8 and input images are cropped into $400 \times 400$ patches for all datasets except the LOLv2-Synthetic subset, for which full-resolution images are used without cropping.

**Evaluation Metrics.** Following our baseline (Yan et al., 2025), for paired datasets, we adopt Peak Signal-to-Noise Ratio (PSNR) and Structural Similarity Index (SSIM) (Wang et al., 2004) as distortion-based metrics to evaluate reconstruction fidelity. To further assess the perceptual quality of the enhanced results, we report the Learned Perceptual Image Patch Similarity (LPIPS) (Zhang et al., 2018), computed using a pretrained AlexNet (Krizhevsky et al., 2012a). For *unpaired* datasets, we employ two no-reference image quality assessment metrics, BRISQUE (Krizhevsky et al., 2012b) and NIQE (Mittal et al., 2012), to evaluate perceptual realism. Moreover, to provide a comprehensive comparison, our method is benchmarked against 11 state-of-the-art supervised learning methods, including RetinexNet (Wei et al., 2018), KinD (Zhang et al., 2019), LLFlow (Wang et al., 2022), EnlightenGAN (Jiang et al., 2021), SNR-Aware (Xu et al., 2022), Bread (Guo & Hu, 2023), PairLIE (Fu et al., 2023), LLFormer (Wang et al., 2023), RetinexFormer (Cai et al., 2023), GSAD (Hou et al., 2023) and CIDNet (Yan et al., 2025), as well as 3 unsupervised learning methods, such as ZeroDCE (Guo et al., 2020), RUAS (Liu et al., 2021) and QuadPrior (Wang et al., 2024) across all datasets.

Table 2: Quantitative result on SID, SICE and the five unpaired datasets (DICM, LIME, MEF, NPE, and VV). The top-ranking score is in **red**.

| Methods | SICE | | SID | | Unpaired | |
|---|---|---|---|---|---|---|
| | PSNR↑ | SSIM↑ | PSNR↑ | SSIM↑ | BRISQUE↓ | NIQE↓ |
| RetinexNet | 12.424 | 0.613 | 15.695 | 0.395 | 23.286 | 4.558 |
| ZeroDCE | 12.452 | 0.639 | 14.087 | 0.090 | 26.343 | 4.763 |
| RUAS | 8.656 | 0.494 | 12.622 | 0.081 | 26.372 | 4.800 |
| LLFlow | 12.737 | 0.617 | 16.226 | 0.367 | 26.087 | 4.221 |
| CIDNet | 13.435 | 0.642 | 22.904 | 0.676 | 23.521 | 3.523 |
| Ours | **15.732** | **0.714** | **23.012** | **0.712** | **21.683** | **3.149** |

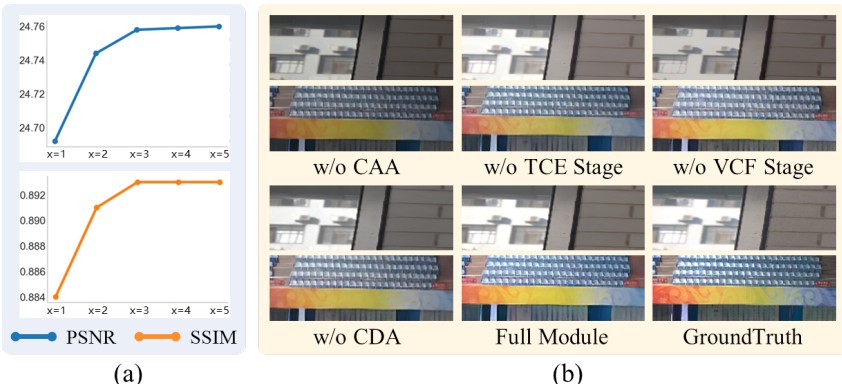

(a)             (b)

Figure 6: (a) shows ablation results for varying quantities of CAA modules, while (b) presents results from experiments using different module types on the LOLv2-REAL dataset.

## 4.2 RESULTS ON PAIRED DATASETS

We evaluate the proposed method on five widely-used paired low-light image enhancement benchmarks: LOLv1, LOLv2(inculding two subsets), SID, and SICE. As shown in Figure 3, our method achieves superior visual quality compared to state-of-the-art approaches. Specifically, RUAS tends to produce over-exposed results that lead to washed-out details, while LLFlow, PairLIE, GSAD, and CIDNet frequently suffer from color distortions, such as hue shifts or unnatural tone mapping. In Figure 5(c), we provide a visual comparison between prior methods and ours. The top row displays the RGB images, while the bottom row shows the corresponding discrete color histograms, offering a more intuitive illustration of the improvements achieved by our approach.

In contrast, our method employs variance-aware channel filtering to selectively suppress noisy or inconsistent features, allowing the model to focus on regions of joint distributional saliency of luminance and color while preserving feature independence to suppress artifacts. This strategy achieving more balanced illumination and faithful color rendition, particularly in challenging low-light regions.

Quantitatively, Table 1 reports the PSNR and SSIM scores across the paired datasets. our method achieves a PSNR of 28.972 on LOLv1, surpassing the best existing method by 0.771 dB. On LOLv2-Real, we obtain a PSNR improvement of 0.647 dB over the previous SOTA, along with an SSIM gain of 0.022. These results demonstrate the effectiveness of our design in producing high-fidelity reconstructions with consistent structure and color. Furthermore, compared with CIDNet, the proposed method introduces only a marginal increase in parameters (+0.08M) and FLOPs (+0.75G), yet consistently outperforms it across multiple benchmark metrics, which validates the efficiency of our design and demonstrates its favorable trade-off between complexity and performance.

## 4.3 RESULTS ON UNPAIRED DATASETS

We assess the generalization ability of models trained on LOLv1 and LOLv2-Synthetic by evaluating them on unpaired low-light datasets using BRISQUE and NIQE, as shown in Table 2. Our method delivers clear performance gains over prior approaches, especially in BRISQUE (a reduction of 1.838). As illustrated in Figure 4 and Figure 8, although RetinexNet achieves competitive BRISQUE scores, our results are visually more realistic and perceptually more natural across diverse scenes. This benefit stems from the color distribution alignment module, which explicitly constrains the chrominance statistics of enhanced images to better match real-world references, improving color fidelity and reducing artifacts common in unpaired enhancement. Figure 8 further shows that our method produces noticeable improvements in challenging cases, particularly in sky regions and in maintaining consistent illumination across the road surface.

## 4.4 ABLATION STUDY

To validate the effectiveness of each component in our VCR framework, we conduct ablations on the LOLv2-real dataset. Table 3 reports results in terms of PSNR, SSIM, and LPIPS. Removing the entire module sig-

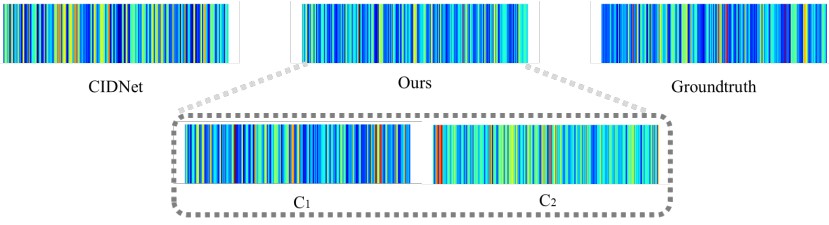

Figure 7: Qualitative comparison of VCF module.

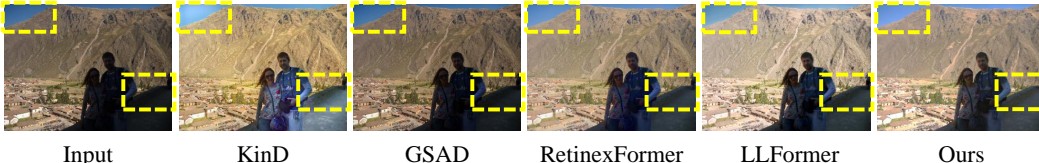

| Input | KinD | GSAD | RetinexFormer | LLFormer | Ours |

Figure 8: Qualitative comparison of enhancement results on the unpaired dataset in one difficult condition.

nificantly reduces performance (23.972 dB PSNR and 0.817 SSIM), confirming the necessity of channel-level recalibration. Excluding the Triplet Channel Enhancement (TCE) stage causes even greater degradation, highlighting the importance of inter-channel and spatial attention. Omitting either Variance-aware Channel Filtering (VCF, $\lambda_{VCF} = 0$) or Color Distribution Alignment (CDA, $\lambda_{CDA} = 0$) also hurts performance, showing their complementary roles in distribution consistency and color fidelity. We further study the effect of stacking multiple CAA modules. As shown in Figure 6(a), increasing the number from $x = 1$ to $x = 2$ consistently improves PSNR and SSIM, while $x = 3$ yields stable but smaller gains. Adding more ($x = 4, 5$) provides negligible improvement but higher cost. Thus, we adopt $x = 3$ in all final experiments for a balance of accuracy and efficiency. Finally, Figure 6(b) shows that removing CDA degrades color fidelity due to the lack of distributional constraints, while eliminating CAA severely impairs both luminance and chrominance quality, underscoring the role of adaptive channel filtering in maintaining illumination balance and accurate color restoration.

Table 3: Ablation studies of modules.

| Exp. | CAA | TCE Stage | VCF ($\lambda_{VCF}$) | CDA ($\lambda_{CDA}$) | PSNR↑ | SSIM↑ | LPIPS↓ |
|---|---|---|---|---|---|---|---|
| 1 | | | | | 23.972 | 0.817 | 0.135 |
| 2 | ✓ | | | | 24.158 | 0.825 | 0.129 |
| 3 | ✓ | ✓ | | | 24.364 | 0.839 | 0.116 |
| 4 | ✓ | ✓ | ✓ | | 24.519 | 0.873 | 0.112 |
| 5 | ✓ | ✓ | ✓ | ✓ | 24.758 | 0.893 | 0.105 |

In addition, several supplementary analyses are also necessary. We also conduct an ablation study on the LOLv2-real dataset to evaluate the contribution of the proposed TCE module (shown in Figure 5(a)). When removing TCE, the model achieves a PSNR of 24.683 and an SSIM of 0.874. Replacing TCE with a vanilla convolution layer of identical parameter size (TCE-Vanilla Conv) yields a PSNR of 24.702 and an SSIM of 0.889. In contrast, our full model equipped with TCE reaches a PSNR of 24.758 and an SSIM of 0.893. These results demonstrate that TCE brings a clear performance gain beyond what can be obtained by simply increasing convolutional capacity, verifying its effectiveness in enhancing feature modeling.

To further validate the effectiveness and distinction of our method compared with CIDnet, we conduct an additional experiment on the LOLv2-real dataset by replacing the HVI space with the HSV space (shown in Figure 5(b)). The results show that in the HVI space, CIDnet achieves a PSNR of 24.111 and an SSIM of 0.871, while our method improves these metrics to 24.758 and 0.893. In the HSV space, CIDnet obtains a PSNR of 21.349 and an SSIM of 0.801, whereas our method increases them to 22.104 and 0.853. These results demonstrate that our approach consistently enhances image quality across both HVI and HSV spaces, indicating strong reliability and generalization capability.

To further verify whether the channels selected by VCF based on covariance variance can effectively reflect feature representation quality, we design a combined qualitative and quantitative evaluation (shown in Figure 7). Four types of inputs are compared against the ground truth by computing feature similarity. CIDnet achieves a similarity of 0.8788 using intensity–color features in the HVI space. In our method, the VCF module separates two subsets of channels: channels with small covariance variance differences (denoted as C1), which achieve a similarity of 0.8965, and channels with large covariance variance differences (denoted as C2), which yield a similarity of 0.8549. Moreover, the overall intensity–color representation produced by our method reaches a similarity of 0.9009. These comparisons clearly indicate that the channels with smaller covariance variance selected by VCF lead to higher-quality feature representations. Visualization results are provided in Fig. 6. In this experiment, we first concatenate the intensity and color-space features, and then apply global average pooling to obtain an HW×1 vector for similarity computation and visual analysis.

## 5 CONCLUSION

We introduce VCR, a novel low-light image enhancement framework designed to improve feature representation in the channel dimension through variance-aware filtering and distribution-level alignment. By enhancing intra-channel consistency between luminance and chrominance via variance-aware filtering and aligning chrominance distributions, our method reduces artifacts and color shifts, resulting in visually natural enhancements. Moreover, VCR achieves state-of-the-art performance on ten public benchmarks, demonstrating superior visual quality and strong generalization across diverse lighting conditions.

## ETHICS STATEMENT

This research adheres to the ICLR Code of Ethics. We ensure that no ethical violations have occurred during the research process. All datasets used comply with publicly available privacy policies, and we have ensured the security and privacy of the data during collection and use. There are no conflicts of interest or funding issues in this research. All methods and applications used in this research follow principles of fairness and objectivity to ensure the integrity and transparency of the research.

## REPRODUCIBILITY STATEMENT

All improvements in this research are based on open-source code and datasets. We provide comprehensive experimental details and algorithm descriptions, including the models, datasets, and training processes used. All relevant source code and datasets will be made open-source. We encourage readers to use the same experimental setups and parameters to reproduce our results and validate the theories and algorithms presented in this work, ensuring the reproducibility of the research and supporting the validation of the results.

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

# A    APPENDIX

This supplementary document provides additional details on the objective evaluation metrics, extended subjective experiments, and further failure cases. We also include implementation and training details to facilitate reproducibility.

## A.1    OBJECTIVE EVALUATION METRICS

In our work, we employ a range of full-reference and no-reference image quality assessment metrics to comprehensively evaluate the performance of low-light image enhancement algorithms. Below, we describe each metric, its underlying principle, and its relevance to our task.

### A.1.1    PEAK SIGNAL-TO-NOISE RATIO (PSNR)

PSNR is one of the most widely used full-reference metrics in image processing. It measures the ratio between the maximum possible pixel intensity value and the mean squared error (MSE) between an enhanced image and its ground-truth reference. Formally, given a ground-truth image $I_{gt}$ and an enhanced image $I_{out}$ of size $H \times W$, the MSE is defined as:

$$\text{MSE} = \frac{1}{HW} \sum_{i=1}^{H} \sum_{j=1}^{W} \big(I_{out}(i,j) - I_{gt}(i,j)\big)^2.$$

PSNR is then computed in decibels (dB) as:

$$\text{PSNR} = 10 \log_{10}\!\left(\frac{L^2}{\text{MSE}}\right),$$

where $L$ is the dynamic range of pixel values (e.g., 255 for 8-bit images). A higher PSNR indicates closer agreement with the reference, corresponding to lower reconstruction error. Although PSNR is simple to compute and provides a quantitative measure of fidelity, it does not always correlate well with perceived visual quality, especially in the presence of structural distortions or color shifts.

### A.1.2 STRUCTURAL SIMILARITY INDEX (SSIM)

SSIM (Wang et al., 2004) was proposed to address the limitations of pixel-wise metrics like PSNR by incorporating human visual system characteristics. SSIM evaluates similarity between two images based on three components: luminance, contrast, and structure. Given two image patches $x$ and $y$, SSIM is defined as

$$\text{SSIM}(x, y) = \big[l(x,y)\big]^\alpha \cdot \big[c(x,y)\big]^\beta \cdot \big[s(x,y)\big]^\gamma,$$

where

$$\begin{cases} l(x,y) = \frac{2\mu_x\mu_y + C_1}{\mu_x^2 + \mu_y^2 + C_1}, \\ c(x,y) = \frac{2\sigma_x\sigma_y + C_2}{\sigma_x^2 + \sigma_y^2 + C_2}, \\ s(x,y) = \frac{\sigma_{xy} + C_3}{\sigma_x\sigma_y + C_3}. \end{cases}$$

Here, $\mu_x$, $\mu_y$ are the mean intensities of $x, y$; $\sigma_x$, $\sigma_y$ are their standard deviations; $\sigma_{xy}$ is the covariance; and $C_1, C_2, C_3$ are small stabilizing constants. Typically, $\alpha = \beta = \gamma = 1$ and $C_3 = C_2/2$. SSIM values range from $-1$ to $1$, with higher values indicating greater structural similarity. In practice, SSIM better captures perceptual quality—preserving edges and textures—compared to PSNR.

### A.1.3 LEARNED PERCEPTUAL IMAGE PATCH SIMILARITY (LPIPS)

LPIPS (Zhang et al., 2018) is a learned perceptual metric that quantifies the perceptual distance between two images by comparing deep features extracted from a pretrained convolutional neural network (e.g., AlexNet, VGG, or SqueezeNet). Given a pair of images, LPIPS computes feature maps at multiple layers $\{F_l(\cdot)\}$ and measures the (normalized) $\ell_2$ distance between these features:

$$\text{LPIPS}(I_{\text{gt}}, I_{\text{out}}) = \sum_l w_l \frac{1}{H_l W_l} \sum_{i,j} \big\| F_l(I_{\text{gt}})_{i,j} - F_l(I_{\text{out}})_{i,j} \big\|_2,$$

where $H_l, W_l$ are the spatial dimensions of the $l$-th feature map, and $w_l$ are learned weights that balance the contribution of each layer. LPIPS correlates well with human judgments of perceptual similarity and is sensitive to semantic-level differences that PSNR and SSIM may overlook.

### A.1.4 BLIND/REFERENCELESS IMAGE SPATIAL QUALITY EVALUATOR (BRISQUE)

BRISQUE (Krizhevsky et al., 2012b) is a no-reference (blind) image quality assessment metric that models natural scene statistics (NSS) in the spatial domain. BRISQUE computes locally normalized luminance coefficients, fits these coefficients to an asymmetric generalized Gaussian distribution (AGGD), and extracts statistical features (e.g., shape and variance parameters). A support vector regressor, trained on human-rated quality scores, maps these features to a quality score. Lower BRISQUE values indicate better perceptual quality. Because it does not require ground-truth references, BRISQUE is particularly useful for evaluating unpaired or real-world images.

### A.1.5 NATURALNESS IMAGE QUALITY EVALUATOR (NIQE)

NIQE (Mittal et al., 2012) is another blind quality metric that evaluates deviations from statistical regularities of natural images. NIQE builds a multivariate Gaussian model over a set of NSS features (mean subtracted contrast normalized coefficients, neighbor pixel products, etc.) extracted from pristine natural images. For a test image, NIQE computes the same features and measures the Mahalanobis distance to the learned Gaussian model:

$$\text{NIQE}(I) = \sqrt{(f - \mu)^\top \Sigma^{-1} (f - \mu)},$$

where $f$ is the feature vector of the test image, and $\mu, \Sigma$ are the mean and covariance of features from natural images. Lower NIQE scores reflect closer adherence to natural image statistics.

## A.2 HVI SPACE

According to the Max-RGB, for each individual pixel $x$, the intensity map of image $I$ can be estimated:

$$I_{\max}(x) = \max_{c \in \{R,G,B\}} I_c(x). \tag{12}$$

Meanwhile, according to the sRGB-HSV transformation, the saturation $s$ of the image can be obtained:

$$s = \begin{cases} 0, & I_{\max} = 0 \\ \dfrac{I_{\max} - \min(I_c)}{I_{\max}}, & I_{\max} \neq 0 \end{cases} \tag{13}$$

and the hue $h$ of the image is formulated as follows:

$$h = \begin{cases} 0, & \text{if } s = 0 \\ \left( \dfrac{I_G - I_B}{I_{\max} - \min(I_c)} \right) \bmod 6, & \text{if } I_{\max} = I_R \\ 2 + \dfrac{I_B - I_R}{I_{\max} - \min(I_c)}, & \text{if } I_{\max} = I_G \\ 4 + \dfrac{I_R - I_G}{I_{\max} - \min(I_c)}, & \text{if } I_{\max} = I_B \end{cases} \tag{14}$$

where $s$ and $h$ correspond to any pixel in the saturation map $S(x)$ and hue map $H(x)$, respectively. Moreover, corresponding to HVT in Figure 2, the horizontal chromaticity component $\hat{H}(x)$ and the vertical component $\hat{V}(x)$ are constructed by polarizing the hue angle from HSV into Cartesian space, defined as:

$$\begin{cases} \hat{H}(x) = C_k(x) \cdot S(x) \cdot \cos\left( \dfrac{\pi H(x)}{3} \right), \\ \hat{V}(x) = C_k(x) \cdot S(x) \cdot \sin\left( \dfrac{\pi H(x)}{3} \right), \end{cases} \tag{15}$$

where $C_k(x)$ is a learnable intensity collapse function defined as:

$$C_k(x) = k \cdot \sqrt{\sin\left( \dfrac{\pi I_{\max}(x)}{2} \right) + \varepsilon}, \tag{16}$$

with $k$ as a trainable parameter and $\varepsilon$ as a small constant (set to $10^{-8}$) for training stability. Moreover, as shown in Fig. 2, the Perceptual-inverse HVI Transformation (PHVIT) is performed to convert the HVI space back to HSV. The hue $H(x)$, saturation $S(x)$, and value $V(x)$ are estimated as:

$$\begin{cases} H(x) = \dfrac{1}{2\pi} \cdot \arctan\left( \dfrac{\hat{v}(x)}{\hat{h}(x)} \right) \bmod 1, \\ S(x) = \alpha_S \cdot \sqrt{\hat{h}^2(x) + \hat{v}^2(x)}, \\ V(x) = \alpha_I \cdot I_{\max}(x), \end{cases} \tag{17}$$

where $\alpha_S$ and $\alpha_I$ are linear scaling parameters that control the output image's saturation and brightness, respectively. The normalized intermediate chromaticity coordinates are computed as:

$$\begin{cases} \hat{h}(x) = \dfrac{\hat{H}(x)}{C_k(x) + \varepsilon}, \\ \hat{v}(x) = \dfrac{\hat{V}(x)}{C_k(x) + \varepsilon}. \end{cases} \tag{18}$$

Table 4: Ablation on the masking ratio of Variance-aware Channel Filtering (VCF) module.

| Metrics | PSNR↑ | SSIM↑ | LPIPS↓ |
|---|---|---|---|
| *Masking Ratio* | | | |
| Ratio=1/5 | 23.997 | 0.853 | 0.141 |
| Ratio=1/4 | 24.516 | 0.872 | 0.126 |
| Ratio=1/3 | **24.758** | **0.893** | **0.105** |
| Ratio=1/2 | 22.963 | 0.804 | 0.207 |

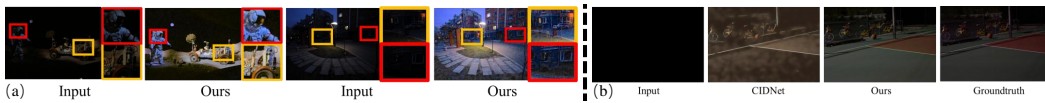

Figure 9: (a) Failure cases of the proposed method under extreme nighttime scenarios. (b) Comparison visualization of SID dataset.

### A.3 VISUAL COMPARISONS

Figure 10, Figure 11 and Figure 12 provides additional visual examples on seven LLIE benchmark datasets, including three paired datasets: LOLv1 (Wei et al., 2018), LOLv2 (Yang et al., 2021), and SICE (Cai et al., 2018), and four unpaired datasets, including DICM (Lv et al., 2018), LIME (Guo et al., 2016), MEF (Ma et al., 2015), NPE (Wang et al., 2013), and VV (Vonikakis et al., 2018). Our method is benchmarked against 11 state-of-the-art supervised learning methods, including RetinexNet (Wei et al., 2018), KinD (Zhang et al., 2019), LLFlow (Wang et al., 2022), EnlightenGAN (Jiang et al., 2021), SNR-Aware (Xu et al., 2022), Bread (Guo & Hu, 2023), PairLIE (Fu et al., 2023), LLFormer (Wang et al., 2023), RetinexFormer (Cai et al., 2023), GSAD (Hou et al., 2023) and CIDNet (Yan et al., 2025), as well as 3 unsupervised learning methods, such as ZeroDCE (Guo et al., 2020), RUAS (Liu et al., 2021) and QuadPrior (Wang et al., 2024) across all datasets. Our method consistently restores both global illumination and local color fidelity, whereas competing methods exhibit over-saturation (e.g., ZeroDCE), residual noise (e.g., LLFlow), or hue shifts (e.g., RetinexNet).

While our VCR framework performs robustly under most conditions, we identify several additional failure scenarios:

High ISO Noise: Images captured with very high ISO exhibit strong sensor noise patterns that our current VCF stage sometimes misinterprets as salient chrominance variation, leading to amplified graininess. Mixed Lighting Sources: Scenes lit by mixed temperature light sources (e.g., tungsten and daylight) can cause uneven color casts. Our CDA module aligns average chrominance distributions but may not fully account for spatially varying color biases.

In future work, we will explore the integration of explicit noise models and adaptive regularization priors into the HVI transform to better handle such extreme conditions.

Figure 9(a) illustrates these failure cases, which we plan to address in future work by incorporating explicit noise priors, enforcing temporal consistency for video data, and adapting color temperature locally. From the visualization in Figure 9(b), we can observe that our method performs significantly better on the SID dataset compared with our baseline (which is trained using the publicly available CIDnet weights).

### A.4 IMPLEMENTATION AND TRAINING DETAILS

#### A.4.1 HYPERPARAMETER SETTINGS

The loss weights are set to $\lambda_{\text{HVI}} = 1$, $\lambda_{\text{VCF}} = 0.2$, and $\lambda_{\text{CDA}} = 0.5$. Temperature for CDA softmax is $\tau = 0.01$. Threshold for VCF masking is $\tau_{\text{VCF}} = 0.1$.

#### A.4.2 TRAINING PROTOCOL

All networks are trained using the Adam optimizer (Kingma & Ba, 2014) ($\beta_1 = 0.9$, $\beta_2 = 0.999$) on a single NVIDIA RTX 3090 GPU. The initial learning rate is $1 \times 10^{-4}$, decayed to $1 \times 10^{-7}$ via cosine annealing over 600 epochs. Batch size is set to 8, and input patches of size $400 \times 400$ are used for all datasets except LOLv2-Synthetic (full resolution). Data augmentation includes random horizontal flips and rotations.

**Mask Ratio in Variance-aware Channel Filtering** We further analyze the impact of the masking ratio in the Variance-aware Channel Filtering (VCF) stage, which determines the proportion of channel covariance entries suppressed by the binary mask $M$. We evaluate four candidate ratios, 1/5, 1/4, 1/3, and 1/2, and report the resulting PSNR and SSIM on the LOLv1 validation set in Table 4. When the ratio is too high, excessive masking removes informative channels and degrades reconstruction fidelity; conversely, a very low ratio fails to adequately suppress distributional inconsistencies. The 1/3 mask ratio strikes the best balance, yielding the highest PSNR of 24.758 dB—an improvement of approximately 0.18 dB over the $\frac{1}{5}$ setting—and the highest SSIM of 0.893. Consequently, we adopt 33% masking as the default configuration for the VCF stage.

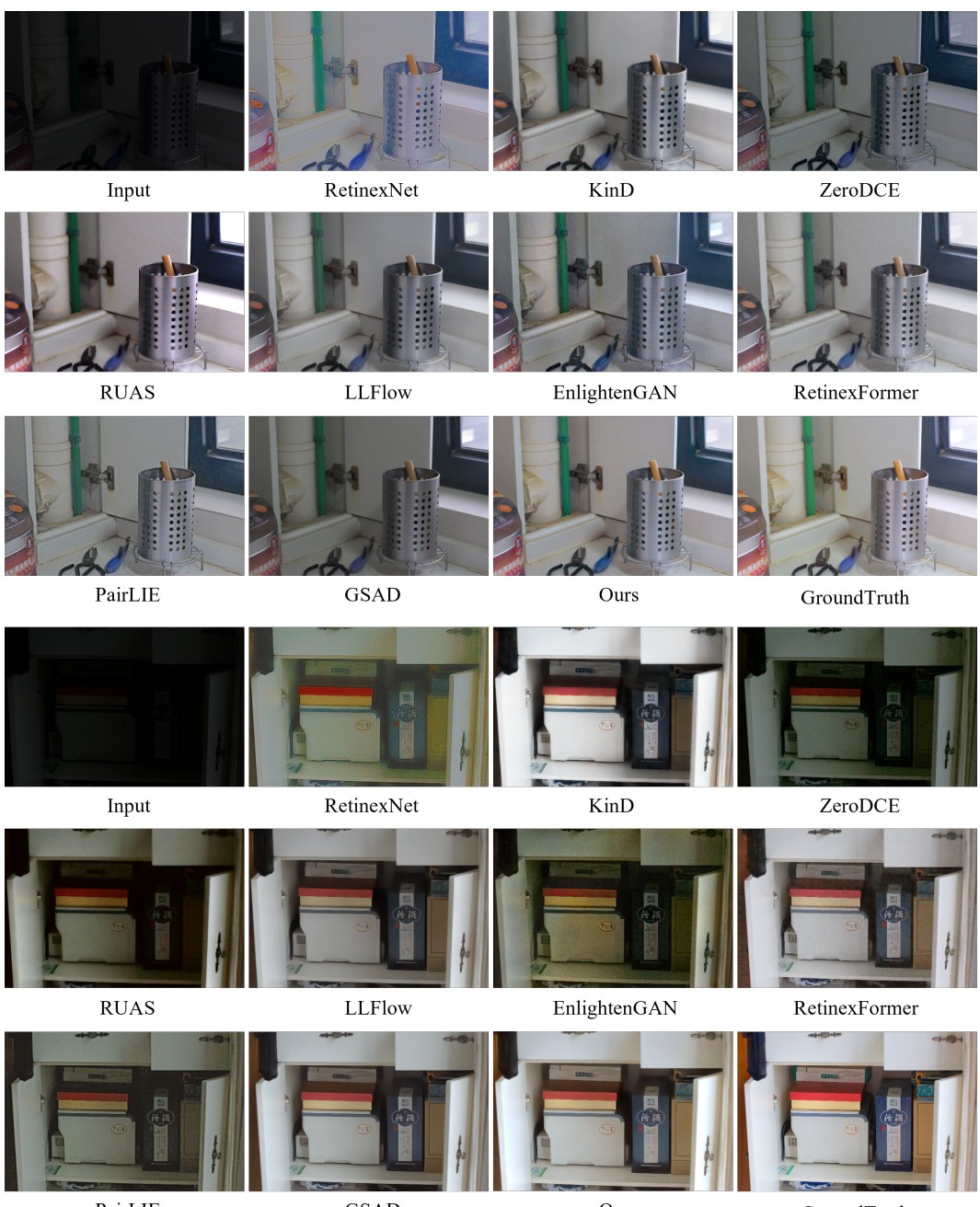

Figure 10: More visual results of various methods on the LOLv1 dataset.

## A.5 Use of Large Models

In this work, large language models are employed solely for language polishing and improving the readability of the manuscript. They are not involved in problem formulation, algorithm design, model implementation, or experimental analysis. All technical contributions and experimental results are independently developed and verified by the authors.

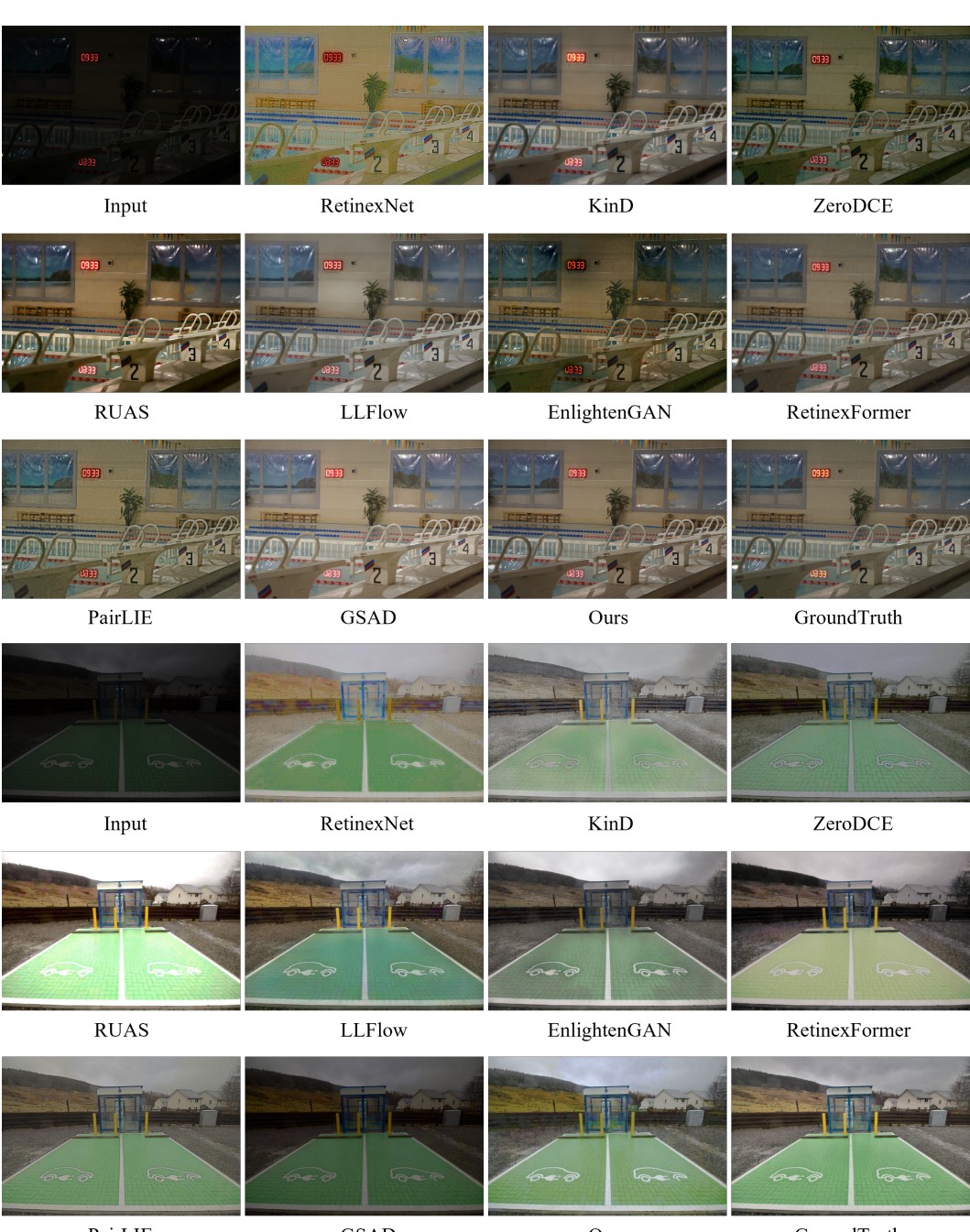

Figure 11: More visual results of various methods on the LOLv2 and SICE dataset.

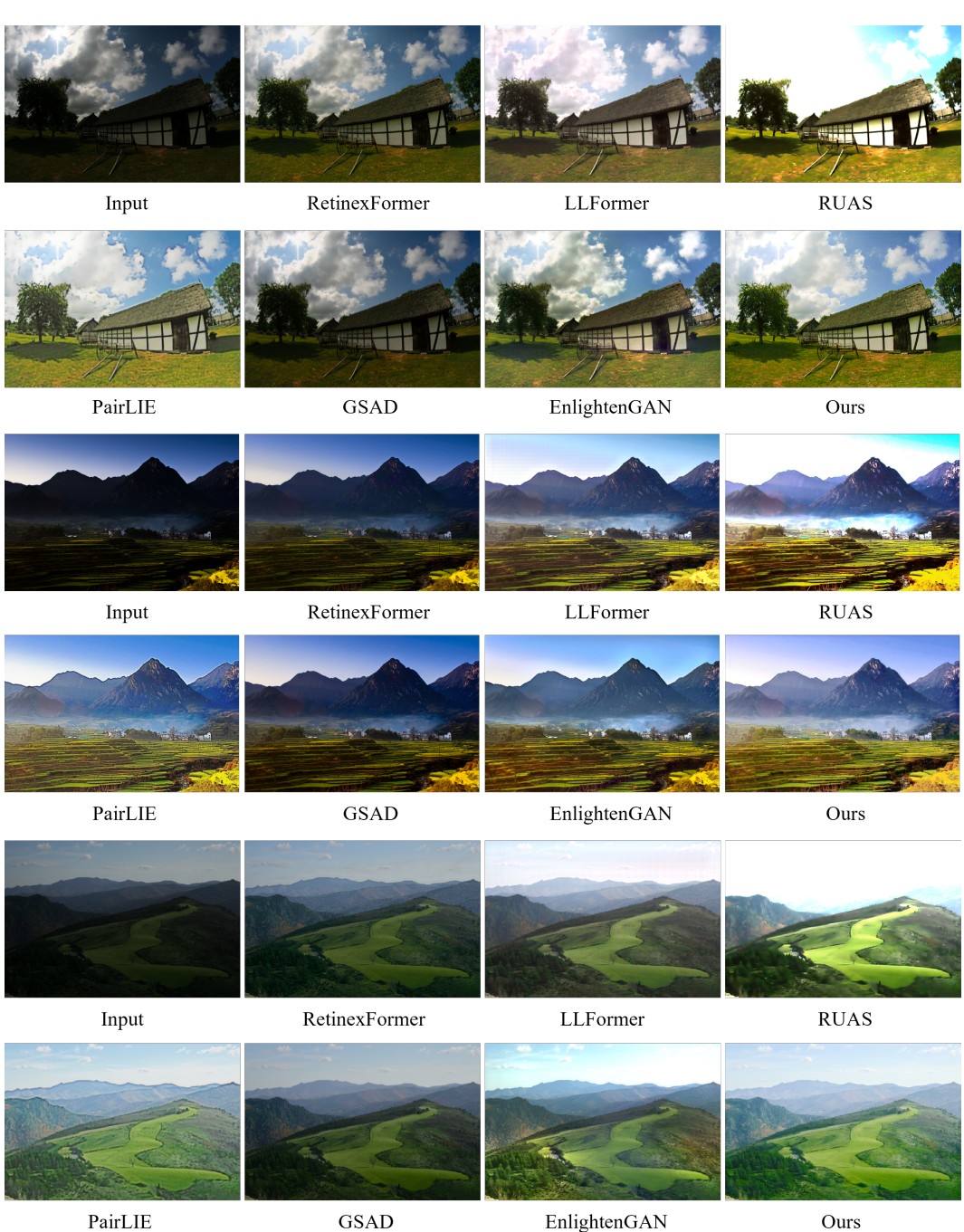

Figure 12: More visual results of various methods on the unpaired dataset.

