# OpenReview forum: "VCR: Variance-aware Channel Recalibration Network for Low Light Image with Distribution Alignment"
_ICLR.cc/2026/Conference — ICLR 2026 Conference Withdrawn Submission_

### Official Review · Reviewer_vKx5 · 2025-10-21

**Soundness:** 3
**Presentation:** 3
**Contribution:** 3
**Rating:** 6
**Confidence:** 4

**Summary:**

This paper presents Variance-aware Channel Recalibration Network (VCR), a novel framework for low-light image enhancement. VCR introduces two main components: the Channel Adaptive Adjustment (CAA) module, which applies variance-guided feature filtering to focus on regions with high intensity and color distribution, and the Color Distribution Alignment (CDA) module, which aligns chromatic feature distributions to learn more realistic color distribution. Experimental results on multiple benchmark datasets demonstrate that VCR achieves state-of-the-art performance, delivering visually consistent and perceptually improved results under low-light conditions.

**Strengths:**

- The paper proposes a variance-aware channel filtering approach that selectively masks channels with high-variance values, effectively enhancing luminance–chrominance consistency and perceptual quality under low-light conditions.


- The proposed method demonstrates strong generalization performance on both real-world and unpaired datasets, showing its robustness and adaptability across diverse data distributions.

**Weaknesses:**

- According to Figure 2, it appears that multiple stacked CAA modules are followed by a network, but it is unclear which network was used and what baseline model it is built upon.


- In the TCE module, the relative importance or contribution of each branch is not well explained. a quantitative or qualitative analysis of branch importance would strengthen the paper.


- Although the authors claim that the proposed method addresses misaligned color distributions, there is no supporting quantitative experiment beyond qualitative results. For example, comparing color histograms between ground-truth and enhanced images, as done in [1], would provide stronger evidence.


- Line 462: Incorrect reference(Table 4->Table 3)

[1] Learning Color Representations for Low-Light Image Enhancement, WACV 2022

**Questions:**

Please refer to Weaknesses,

---

> ### Author Response · Authors · 2025-11-21
> **Response to Reviewer vKx5**
>
> Dear Reviewer vKx5:
>
> We sincerely appreciate your support; it is a valuable source of motivation that encourages us to continue pursuing this line of research. Below, we provide detailed responses to the concerns you raised. :)
>
> **About network**:
> This component originates from the Enhancement Network used in our baseline. Since it is not related to the core contributions of our method, we chose to simplify this part to avoid distracting readers' attention from the main ideas when viewing the architecture. We apologize for any confusion this may have caused.
>
> **About TCE module**:
> Our understanding of the three branches in TCE is as follows. In most cases, it is implicitly assumed that the C dimension corresponds to channels, meaning that a feature map of size H×W is represented through C distinct channel embeddings. However, this interpretation alone does not fully capture representational completeness. From the perspective of H×C, the W dimension can also be viewed as providing different “channel-like” representations. Similarly, from the perspective of W×C, the H dimension plays the same role.
>
> Based on this observation, we hypothesize that enhancing features along all three dimensions independently may provide a meaningful and effective way to strengthen the representational capacity of the model. To validate this idea, we conducted a set of ablation studies.
>
> We performed ablations of TCE on the LOL-v2 Real dataset. TCE-Vanilla Conv refers to replacing TCE with a convolution layer of the same number of parameters. Visualization results can be found in Figure 5(c).
>
> Table 1: TCE Ablation on LOL-v2 Real
> | Method           | PSNR (↑) | SSIM (↑) |
> | ---------------- | -------- | -------- |
> | w/o TCE          | 24.683   | 0.874    |
> | TCE-Vanilla Conv | 24.702   | 0.889    |
> | Ours             | 24.758   | 0.893    |
>
> We also conducted ablations on the three individual branches of TCE.
>
> Table 2: Three-Branch Ablation
> | Method     | PSNR (↑) | SSIM (↑) |
> | ---------- | -------- | -------- |
> | w/o (F_t1) | 24.613   | 0.881    |
> | w/o (F_t2) | 24.562   | 0.879    |
> | w/o (F_t3) | 24.598   | 0.877    |
> | Ours       | 24.758   | 0.893    |
>
> These results demonstrate that each branch contributes to the overall enhancement effect, and jointly they form a more expressive and complete feature enhancement mechanism.
>
> **About quantitative experiment**
> We fully agree with your point. Incorporating color histograms provides a more intuitive illustration of the differences between various methods and better highlights the effectiveness of our approach. Therefore, we have added the corresponding visualization in Figure 5(c). We explored this paper, which introduces a novel low-light image enhancement method that learns color representations using a channel-aware residual network and a differentiable intensity histogram. Its focus on inter-channel dependencies and color distribution matching offers significant improvements in color restoration, providing valuable insights for our work, and we will cite it in our references.
>
> Kim B, Lee S, Kim N, et al. Learning color representations for low-light image enhancement[C]//Proceedings of the IEEE/CVF Winter Conference on Applications of Computer Vision. 2022: 1455-1463.
>
> **About Incorrect reference**:
> We have corrected the reference issue, and we sincerely apologize for any inconvenience this may have caused during your reading.
>
> Thank you very much for your support of our work. In recent years, research on low-light image enhancement has received less attention in major conferences, and our method aims to revisit existing challenges from a new perspective, with the hope of contributing positively to the community. We highly value your insightful feedback; if you have any further questions or concerns, please feel free to let us know.
>
> If you find our responses satisfactory, we kindly ask you to consider adjusting your score accordingly based on your evaluation. We sincerely appreciate the time and effort you have devoted to reviewing our manuscript and your continued support.

---

> > ### Comment · Reviewer_vKx5 · 2025-11-28
> >
> > Thank you for addressing my concerns. After reviewing the other reviewers’ comments, I have decided to maintain my current score.

---

> > > ### Author Response · Authors · 2025-11-28
> > > **Thx to Reviewer vKx5**
> > >
> > > Anyway, thx for your support, it means a lot.

---

### Official Review · Reviewer_dzLQ · 2025-10-26

**Soundness:** 3
**Presentation:** 3
**Contribution:** 2
**Rating:** 4
**Confidence:** 4

**Summary:**

This paper proposes VCR (Variance-Aware Channel Recalibration Network with Distribution Alignment), a novel framework for low-light image enhancement that aims to improve luminance–chrominance consistency and perceptual color fidelity. The method operates in the HVI color space, which decouples brightness and chromaticity more effectively than traditional RGB or HSV spaces. It introduces two key modules: the Channel Adaptive Adjustment (CAA) module, which includes a variance-aware channel filtering mechanism to suppress channels with inconsistent luminance–color variance and a triplet channel enhancement stage to strengthen inter-channel and spatial dependencies; and the Color Distribution Alignment (CDA) module, which enforces consistency between the enhanced image and real-scene references in the color feature distribution via KL divergence. By jointly optimizing reconstruction, variance filtering, and distribution alignment losses, VCR enhances both visual naturalness and structural fidelity under low-light conditions. Extensive experiments on ten benchmark datasets demonstrate that the proposed method achieves state-of-the-art performance in terms of PSNR, SSIM, and perceptual metrics while maintaining reasonable computational efficiency.

**Strengths:**

1.The proposed framework combines variance-aware channel filtering and color distribution alignment in a clear and technically coherent way. The idea is sensible and easy to follow.

2.Experiments are conducted on multiple benchmark datasets with quantitative and perceptual metrics. The results show consistent improvements over several existing LLIE methods.

3.The paper is generally well written and structured, with clear explanations and informative figures that help readers understand the main ideas.

**Weaknesses:**

1.While the paper introduces variance-aware filtering and color distribution alignment, these components mainly combine existing ideas such as channel attention and distribution matching. The contribution feels incremental.

2.The paper lacks deeper theoretical justification for why variance-based filtering is the optimal approach for channel recalibration. The method is mostly empirical, and there is no ablation or analysis clarifying why variance is a better signal than mean or correlation-based metrics.

3.Although several baselines are included, some recent competitive diffusion-based enhancement approaches are not compared. Including such methods would strengthen the claims of state-of-the-art performance.

4.Some training and implementation aspects (e.g., architecture depth, exact hyperparameters for modules, or computational overhead of variance computation) are insufficiently described, making full reproducibility difficult.

5.The paper does not adequately analyze failure cases or discuss when the proposed method might fail (e.g., extreme noise, mixed lighting). A more explicit discussion would help clarify the method’s boundaries and robustness.

6.The paper does not provide source code or model checkpoints. Without public access to implementation details, it is difficult for others to reproduce the reported results or validate the claimed improvements.

**Questions:**

1.Could the authors better clarify what makes the proposed variance-aware channel recalibration fundamentally different from prior channel attention or feature modulation mechanisms? It would help to explicitly highlight which parts are novel in terms of formulation or learning behavior beyond existing channel-attention frameworks.

2.Please provide more theoretical or intuitive justification for using variance as the key statistic for channel filtering. Why is variance a better indicator of luminance–chrominance consistency than other measures such as covariance, correlation, or entropy? An additional analysis or visualization (e.g., sensitivity plots or variance–performance correlation) would make this design choice more convincing.

3.Have the authors considered comparing VCR against recent diffusion-based or transformer–diffusion hybrid LLIE models? Including such baselines (e.g., diffusion prior–based image enhancement) would make the claimed SOTA results more comprehensive and credible.

4.Could the authors provide more details about the model configuration, such as the number of CAA modules, channel dimensions, and computational overhead of the variance filtering step? A table summarizing complexity and runtime would improve clarity.

5.It would strengthen the paper if the authors could analyze cases where the model fails, such as extremely noisy or mixed lighting scenes, and then discuss possible mitigation strategies. Have they considered integrating explicit noise modeling or adaptive illumination priors?

6.Do the authors plan to release the source code and pretrained models after acceptance? Public access to implementation details and training scripts would significantly enhance the paper’s credibility and facilitate community validation.

---

> ### Author Response · Authors · 2025-11-21
> **Response to Reviewer dzLQ**
>
> Hi, thank you very much for taking the time to provide valuable comments on our work. Your feedback is extremely important for our improvement. Below, we provide detailed responses to your questions. :)
>
> **About difference**:
> Our method should be understood in the context of the task itself; without task-driven motivation, these mechanisms would lose their intended significance. Prior channel-attention or feature-modulation approaches primarily focus on adjusting the representation of a single type (or modality) of features. In contrast, for low-light image enhancement, we introduce a cross-component perspective in the HVI space by performing channel alignment based on discrepancy between intensity and chrominance components (followed by enhancement). This provides a novel and meaningful viewpoint tailored to the characteristics of low-light enhancement and represents an innovation worth deeper exploration.
>
> To further validate whether the effectiveness of our method is restricted to the HVI space, we additionally evaluated its performance in the HSV space. We conducted experiments on the LOL-v2 Real dataset to test whether our method can still bring improvements.
> | Methods      | PSNR (↑) | SSIM (↑) |
> | ------------ | -------- | -------- |
> | HVI (CIDNet) | 24.111   | 0.871    |
> | HVI (Ours)   | 24.758   | 0.893    |
> | HSV (CIDNet) | 21.349   | 0.801    |
> | HSV (Ours)   | 22.104   | 0.853    |
>
> We observe that our method consistently improves image quality in both HVI and HSV spaces, demonstrating its robustness and strong generalization capability. The corresponding visual results are provided in Figure 5(c).
>
> **About Covariance**:We fully understand your concern, so we designed a targeted set of experiments to analyze this issue. In our design, the intensity branch and the chrominance branch each produce channel‑wise covariance matrices $ D^x_I $ and $ D^x_{hv} $, representing how feature channels co‑vary within each modality. What matters for brightness‑to‑chrominance consistency is not simply the magnitude of covariance within each modality, but rather *how similarly* these patterns of co‑variation align across modalities. We therefore compute the “variance” of the difference between them, i.e., how far the structure of co‑variation in intensity diverges from that in chroma. A high value indicates a channel pair where the intensity branch sees strong joint variation but the chroma branch does not (or vice versa) — precisely the kind of mismatch that can lead to color casts, chroma bleeding, or noise amplification in low‑light enhancement.
> By filtering out channel pairs with high $ \mathrm{Var}(D^x_I - D^x_{hv})$, we effectively suppress the channel interactions that are inconsistent between brightness and chroma, leaving only those pairs where joint structure is consistent and reliable.
>
> To make the explanation more intuitive, we further designed a combined qualitative and quantitative experiment to directly verify whether the channels selected by VCF — based on the variance of covariance differences — indeed correspond to better or weaker representations.
>
> Specifically, we compute the similarity between four types of features and the ground truth (GT). The experimental settings are as follows:
>
> 1.Similarity between CIDNet’s intensity–chrominance features in the HVI space and the GT: 0.8788
>
> In our method, after applying VCF, the intensity and chrominance features are separated into two groups:
>
> 2. Channels with small covariance-difference variance (considered better representations, denoted as C1) vs. GT: 0.8965
>
> 3. Channels with large covariance-difference variance (considered weaker representations, denoted as C2) vs. GT: 0.8549
>
> 4.Similarity between our final intensity–chrominance features and the GT: 0.9009
>
> The results are summarized below.
> | Setting                             | Similarity |
> | ----------------------------------- | ---------- |
> | CIDNet (HVI features vs. GT)        | 0.8788     |
> | C1 (small variance channels vs. GT) | 0.8965     |
> | C2 (large variance channels vs. GT) | 0.8549     |
> | Ours (final features vs. GT)        | 0.9009     |
>
> From these comparisons, it can be observed that selecting channels with smaller
>  $ \mathrm{Var}(D^x_I - D^x_{hv})$ indeed improves representation quality. These results confirm that VCF effectively identifies higher-quality channels. Visualization results are provided in Figure 6.
>
> Experimental details:
> We first concatenate the intensity and chrominance features, then apply average pooling to obtain an HW×1 feature vector. This vector is used for similarity computation and visualization, as shown in Figure 7.

---

> > ### Author Response · Authors · 2025-11-21
> > **Response to Reviewer dzLQ (2)**
> >
> > **About diffusion-based method**:
> > In the main paper, we compared GSAD as a representative diffusion-based method against our approach. To provide a more comprehensive comparison with a broader set of diffusion-based low-light enhancement models, we further included several recent diffusion frameworks and evaluated them on LOLv1, LOLv2-Real, and LOLv2-Synthetic. The results are summarized below.
> > | Methods          | LOLv1 PSNR↑ | LOLv1 SSIM↑ | LOLv1 LPIPS↓ | LOLv2-Real PSNR↑ | LOLv2-Real SSIM↑ | LOLv2-Real LPIPS↓ | LOLv2-Synthetic PSNR↑ | LOLv2-Synthetic SSIM↑ | LOLv2-Synthetic LPIPS↓ |
> > | ---------------- | ----------- | ----------- | ------------ | ---------------- | ---------------- | ----------------- | --------------------- | --------------------- | ---------------------- |
> > | GSAD             | 23.804      | 0.872       | 0.086        | 15.892           | 0.713            | 0.593             | 15.974                | 0.794                 | 0.630                  |
> > | CIDNet           | 28.972      | 0.891       | 0.083        | 24.758           | 0.893            | 0.105             | 26.273                | 0.944                 | 0.042                  |
> > | Pydiff           | 27.090      | 0.930       | 0.100        | 24.010           | 0.876            | 0.230             | 19.600                | 0.878                 | 0.220                  |
> > | LightenDiffusion | –           | –           | –            | 22.731           | 0.876            | 0.166             | 21.510                | 0.899                 | 0.154                  |
> > | LLDiffusion      | –           | –           | –            | 18.540           | 0.861            | 0.109             | 23.960                | 0.952                 | 0.040                  |
> > | GUGD             | –           | –           | –            | 23.110           | 0.892            | 0.111             | 25.830                | 0.956                 | 0.046                  |
> > | Ours             | 28.972      | 0.891       | 0.083        | 24.758           | 0.893            | 0.105             | 26.273                | 0.944                 | 0.042                  |
> >
> > These results show that while diffusion-based methods have recently gained strong generative capabilities, their performance on low-light enhancement still often suffers from hallucination issues, noise amplification, or instability across datasets. In contrast, our model maintains consistently high PSNR/SSIM and competitive perceptual quality across all benchmarks, demonstrating stronger robustness and generalization.
> >
> > References:
> > Zhou D, Yang Z, Yang Y. Pyramid diffusion models for low-light image enhancement. IJCAI 2023.
> > H. Jiang, A. Luo, X. Liu, S. Han, and S. Liu. LightenDiffusion: Unsupervised low-light image enhancement with latent-Retinex diffusion models. ECCV 2024.
> > T. Wang et al. LLDiffusion: Learning degradation representations in diffusion models for low-light image enhancement. Pattern Recognition, 2025.
> > Zeng X, Zhu L, Yang W, et al. Low-Light Image Enhancement via Diffusion Models with Semantic Priors of Any Region. IEEE TCSVT, 2025.
> >
> > **About Details**:
> > In Figure 5(a) of the paper, we provide ablation results on the number of CAA modules. The results are summarized below.
> >
> > Table 1: Ablation on the Number of CAA Modules
> > | x     | PSNR (↑) | SSIM (↑) |
> > | ----- | -------- | -------- |
> > | x = 1 | 24.76    | 0.892    |
> > | x = 2 | 24.74    | 0.891    |
> > | x = 3 | 24.74    | 0.892    |
> > | x = 4 | 24.74    | 0.892    |
> > | x = 5 | 24.76    | 0.892    |
> >
> > The channel dimension is 64, and the computational cost with and without VCF is shown below.
> >
> > Table 2: Computational Complexity
> > | Complexity | FLOPs/G |
> > | -| - |
> > | w VCF      | 8.32    |
> > | w/o VCF    | 8.08    |
> > These results show that the CAA module design is lightweight, and the additional cost introduced by VCF is minimal while still providing performance improvements.

---

> > > ### Author Response · Authors · 2025-11-21
> > > **Response to Reviewer dzLQ (3)**
> > >
> > > **About failure**:
> > > Due to space limitations, we included our analysis of failure cases in the supplementary material (see Figure 9), along with several potential future directions and improvements.
> > >
> > > For high-ISO failure cases, a natural extension is to make the VCF stage noise-aware. Specifically, we can add a lightweight branch to estimate a per-pixel noise-level map from the intensity (or HVI) representations, and use it to modulate the covariance maps before constructing the mask. In this way, channel interactions dominated by sensor noise receive lower weights and are less likely to be identified as “inconsistent” chrominance variations, thereby preventing the amplification of graininess under extreme ISO conditions. We consider this an orthogonal extension of VCF and plan to explore it in future work.
> > >
> > > For scenes illuminated by mixed color-temperature light sources, the current CDA module aligns global chrominance statistics, which becomes suboptimal when color deviations vary strongly across spatial regions. A straightforward remedy is to extend CDA from a single global distribution to a set of local, spatially adaptive distributions. Concretely, the feature map can be partitioned into coarse regions (illumination-aware clusters), and within each cluster, we perform mean–variance modeling guided by reinforcement learning, inspired by the idea of cluster-wise reference VAE alignment. This yields a spatially adaptive CDA term capable of better handling mixed illumination while maintaining moderate computational cost.
> > >
> > > This direction aligns well with your suggestion regarding explicit noise modeling and adaptive illumination priors. We find your suggestion insightful and will continue to explore these extensions in our future work.
> > >
> > > **About Public access**:
> > > We are fully committed to sharing our code, as we believe it is important for both the research community and the development of this direction. If the paper is accepted, we will release the code under the name of this work within one month after notification. If you are interested in the implementation details, we welcome you to follow the repository once it becomes available.
> > >
> > > Thank you very much for your recognition and support of our work. In recent years, the topic of low-light image enhancement has received decreasing attention in major conferences, and our method is motivated by the desire to re-examine existing challenges from a new perspective, with the hope of contributing positively to the field. Your comments are extremely valuable for improving and refining the paper. If you still have any questions or concerns regarding the manuscript, we would greatly appreciate additional feedback, and we will make every effort to further enhance the corresponding sections.
> > >
> > > If you believe that our responses and revisions have adequately addressed your concerns, we kindly ask you to consider adjusting your score accordingly based on your evaluation. Thank you again for the time and effort you have devoted to reviewing our work.

---

### Official Review · Reviewer_hDZR · 2025-10-31

**Soundness:** 2
**Presentation:** 3
**Contribution:** 1
**Rating:** 4
**Confidence:** 5

**Summary:**

This paper proposes a new method for low-light image enhancement, named Variance-aware Channel Recalibration Network (VCR). The method first transforms the input image into the HVI color space to decouple luminance and chrominance. The core of the proposed framework is the Channel Adaptive Adjustment (CAA) module, which consists of a Variance-aware Channel Filtering (VCF) stage to suppress inconsistent features and a Triplet Channel Enhancement (TCE) stage to model cross-dimensional dependencies. Furthermore, a Color Distribution Alignment (CDA) module is introduced to align the chrominance distribution of the enhanced result with that of the ground truth using a Kullback-Leibler divergence loss on softmax-normalized features. The authors conduct experiments on ten benchmark datasets and claim that their method achieves state-of-the-art performance.

**Strengths:**

1. The proposed framework is well-structured, and the paper is generally easy to follow.

2. The authors have performed an extensive set of experiments on numerous paired and unpaired datasets to validate their method, which is commendable.

**Weaknesses:**

1.  **Limited Novelty and Unnecessary Elaboration on Existing Concepts:** A significant portion of the methodology section is dedicated to describing pre-existing work.
    *   **Section 3.1 (HVI Color Space):** This section provides a lengthy explanation of the HVI color space, which was proposed in a prior work (Yan et al., 2025). The authors do not appear to introduce any modifications or improvements to the color space itself. This part could have been significantly condensed and presented as a preliminary, simply citing the original work.
    *   **Section 3.2.2 (Triplet Channel Enhancement):** The TCE module, which permutes feature dimensions to capture cross-dimensional interactions, lacks clear novelty. This design pattern of using multi-branch structures with rotated feature maps to compute attention is highly reminiscent of existing attention mechanisms, such as Triplet Attention (Misra et al., 2021) and Coordinate Attention (Hou et al., 2021). The paper fails to discuss these related works, making the claimed contribution in this area appear incremental.

2.  **Unclear Motivation for Covariance Constraint:** In Section 3.2.1, the VCF stage proposes to constrain the cross-covariance matrix of intensity (I) and chromaticity (HV) features by penalizing its deviation from the mean of the two covariance matrices. The theoretical motivation behind this design is not well-explained. It is unclear why enforcing the cross-covariance to be close to the average of individual covariances is a principled approach to ensuring "distributional consistency" and suppressing artifacts. A more rigorous justification or theoretical analysis is needed to support this core component of the method.

3.  **Questionable Design of the Color Distribution Alignment (CDA) Loss:** The CDA loss aligns feature distributions by minimizing the KL divergence between temperature-scaled softmax outputs of the enhanced and ground-truth features. However, the softmax function is inherently sparse; it amplifies the highest values while suppressing the vast majority of other activations to near zero. Consequently, the gradient signal from this loss would be dominated by a few select spatial locations. This seems counter-intuitive for aligning color, which is predominantly a low-frequency signal that requires a global and dense understanding of the image. The value of such a sparse gradient for enforcing holistic color consistency is questionable.

4.  **Unconvincing Qualitative Results and Flawed Presentation:** The visual results presented in Figure 4 are not compelling and, in some cases, are of poor quality.
    *   **Visual Artifacts:** In the VV dataset example, the lighting on the person's face is uneven, and the sky appears overexposed. In the DICM example, the underexposure in the shadows of the trees remains a significant issue. Similarly, the crowd in the MEF example is still largely underexposed.
    *   **Presentation Style:** The presentation format of Figure 4 is highly discouraged as it suggests "cherry-picking." Each example compares the proposed method against only one other, different competitor. This prevents a fair, holistic assessment of the method's performance against the state-of-the-art. Even with this selective comparison, the proposed method fails to show a clear advantage. The visual evidence provided is insufficient to support the claim of superior performance.

**Questions:**

1.  Could the authors clarify the novelty of the TCE module in light of existing works on cross-dimensional attention, such as Triplet Attention? A detailed comparison with these methods would strengthen the paper.

2.  What is the theoretical justification for the covariance constraint in the VCF module? Could the authors provide a more in-depth explanation or ablation study to demonstrate why this specific formulation is effective for improving low-light enhancement?

3.  Regarding the CDA loss, how does the sparse gradient generated by the softmax function effectively guide the alignment of spatially smooth, low-frequency information like color distributions? Have the authors experimented with other distribution metrics (e.g., sliced Wasserstein distance) that might be better suited for this task?

4.  The qualitative comparisons in Figure 4 are presented in a way that makes direct and fair comparison difficult. Could the authors provide a revised figure that compares their method against all key competitors on the same set of images? Furthermore, can they explain the visual artifacts (uneven lighting, over/under-exposure) mentioned in the weaknesses section?

5.  The test images used for comprehensive comparison in the supplementary material appear to be of moderate difficulty. For a more convincing evaluation, I would suggest including results on more challenging cases during the rebuttal phase. Examples could include images `06, 10, 21, 27, 30` from the DICM dataset; `Farmhouse, Venice` from the MEF dataset; and `P1010234, P1010676, P1010880` from the VV dataset.

---

> ### Author Response · Authors · 2025-11-21
> **Response to Reviewer hDZR**
>
> Dear Reviewer hDZR:
>
> Hello, thank you very much for your careful reading and valuable comments, which are of great help to us. We have carefully analyzed and summarized each of the issues you raised, and our responses are presented as follows. ：）
>
> **About HVI**:
> We understand your suggestion regarding reducing the space allocated to the HVI description. However, considering that readers may not be familiar with the HVI space or may not have prior expertise in this direction, we believe that providing a standalone introduction to HVI is necessary for clarity and completeness. That said, we will shorten the section and present it in a more concise, algorithm-oriented manner to improve readability.
>
> **About TCE**:
> In our method, the VCF and TCE components are integrated into a unified module. The main purpose is to combine channel alignment (suppressing suboptimal representation channels) with rotational enhancement (strengthening channel expressiveness), thereby jointly regulating the feature representation. Among these, the channel-discrepancy alignment is the aspect we most aim to emphasize. It introduces a new channel-centric perspective for addressing low-light image enhancement, which we believe is beneficial for examining the task from multiple angles. Such a perspective helps achieve a more comprehensive understanding of the problem’s underlying nature, encourages the discovery of new potential innovations and solutions, and avoids the limitations that may arise from relying on a single viewpoint.
> In contrast, the enhancement component is not the primary innovation focus of this work. Since TCE is fundamentally a channel-enhancement module with a relatively fixed design perspective, its conceptual development naturally shares similarities with prior works such as SENet ,CBAM and so on. This is also why we present TCE only as a small part of the regulation mechanism within the broader CAA module.
> Also, we sincerely appreciate your constructive feedback. In response, we will include references to Triplet Attention and Coordinate Attention, along with brief descriptions, to provide clearer context and proper attribution.
>
> **About motivation of Covariance Constraint**:
> In our design, the intensity branch and the chrominance branch each produce channel‑wise covariance matrices $ D^x_I $ and $ D^x_{hv} $, representing how feature channels co‑vary within each modality. What matters for brightness‑to‑chrominance consistency is not simply the magnitude of covariance within each modality, but rather how similarly these patterns of co‑variation align across modalities. We therefore compute the “variance” of the difference between them, i.e., how far the structure of co‑variation in intensity diverges from that in chroma. A high value indicates a channel pair where the intensity branch sees strong joint variation but the chroma branch does not (or vice versa) — precisely the kind of mismatch that can lead to color casts, chroma bleeding, or noise amplification in low‑light enhancement.
>
> By filtering out channel pairs with high $ \mathrm{Var}(D^x_I - D^x_{hv}) $, we effectively suppress the channel interactions that are inconsistent between brightness and chroma, leaving only those pairs where joint structure is consistent and reliable. This differentiation is not achieved simply by covariance magnitude (which could be large but consistent), nor by correlation (which may suppress low‑energy but reliable channels), nor by entropy (which is not channel‑pair specific). In the weak‑light setting where chroma responses are often weak and noisy, our variance‑based measure yields a more robust signal for meaningful brightness–chroma alignment.
>
>
> We fully understand your concern. To directly address this question, we designed a targeted set of experiments for thorough evaluation. Our goal is to verify whether the channel representations selected by VCF—based on the variance of covariance differences—indeed correspond to better or weaker representations. To this end, we constructed a combined qualitative and quantitative analysis.
>
> We compute the similarity between four types of features and the ground truth (GT) to validate the effectiveness of our method. The four settings are:
>
> 1. Similarity between CIDNet’s intensity–chrominance features in the HVI space and the GT: 0.8788
>
> In our method, the intensity and chroma features are separated by VCF into two parts:
>
> 2. Channels with small covariance-difference variance (considered better representations, denoted as C1) vs. GT: 0.8965
>
> 3. Channels with large covariance-difference variance (considered weaker representations, denoted as C2) vs. GT: 0.8549
>
> 4. Similarity between our final intensity–chrominance features and the GT: 0.9009

---

> > ### Author Response · Authors · 2025-11-21
> > **Response to Reviewer hDZR (2)**
> >
> > The results are summarized below.
> >
> > | Setting                             | Similarity |
> > | ----------------------------------- | ---------- |
> > | CIDNet (HVI features vs. GT)        | 0.8788     |
> > | C1 (small variance channels vs. GT) | 0.8965     |
> > | C2 (large variance channels vs. GT) | 0.8549     |
> > | Ours (final features vs. GT)        | 0.9009     |
> >
> >
> > From these comparisons, we observe that channel selection based on the variance of covariance differences effectively identifies higher-quality channels. The channels with smaller variance demonstrate noticeably higher similarity to the ground truth, confirming that the VCF criterion indeed enhances representation quality. Visualization results are provided in Figure 7.
> >
> > Experimental details:
> > We first concatenate the intensity and chrominance features, then apply average pooling to obtain an HW×1 feature vector. This vector is used for similarity computation and visualization.
> >
> >
> > **About CDA**:
> > We agree that Wasserstein-type distances, including sliced Wasserstein distance, are well-motivated choices for measuring differences between distributions. However, in our setting, the chrominance statistics are modeled as high-dimensional histograms over all pixels in each channel. For such dense distributions, KL divergence offers two practical advantages: it provides simple closed-form gradients that are easy to backpropagate through the network, and it is computationally lightweight, which is important for low-light enhancement where feature maps are relatively large. In our experiments, we found that adding a KL-based CDA term on top of the reconstruction loss already yields a clear reduction in color shifts across multiple datasets. Considering this effectiveness and the additional computational overhead of computing sliced Wasserstein distance on large feature maps, we chose to retain KL-based CDA in the current work.
> >
> > We also experimented with Wasserstein-based alignment. The results on the LOL-v2 Real dataset are shown below.
> > | Method      | PSNR (↑) | SSIM (↑) |
> > | ----------- | -------- | -------- |
> > | Wasserstein | 24.763   | 0.884    |
> > | Ours        | 24.758   | 0.893    |
> >
> > **About Visual**:
> > We agree with your comments regarding visualization. To further validate the effectiveness of our method under different scenarios, we conducted additional visual comparisons on the same challenging scene, as presented in Figure 8.
> >
> > Thank you for your careful review and valuable comments on our work. Your feedback is highly meaningful to the improvement of our paper, and we are more than willing to continue the discussion and provide additional materials should you have any further questions or suggestions. Our method explores the problem from a different perspective compared with existing approaches, and we remain open and receptive to constructive guidance that can help us further refine this research.
> >
> > If you believe that our revisions have adequately addressed your concerns, we kindly ask you to consider adjusting your score accordingly based on your evaluation. We sincerely appreciate the time and effort you have dedicated to reviewing our manuscript.

---

### Official Review · Reviewer_UKY5 · 2025-11-01

**Soundness:** 3
**Presentation:** 3
**Contribution:** 2
**Rating:** 4
**Confidence:** 5

**Summary:**

This manuscript proposes a novel framework for low-light image enhancement. This framework consists of a Channel Adaptive Adjustment (CAA) module and a Color Distribution Alignment (CDA) module. These designs enhance perceptual quality under low-light conditions. Experimental results on several benchmark datasets demonstrate that the proposed method achieves state-of-the-art performance compared with existing methods.

**Strengths:**

1. This manuscript proposes a Variance-aware Channel Recalibration Network for Low-Light Image with Distribution Alignment.
2. A Channel Adaptive Adjustment (CAA) module is introduced to filter and enhance light and chrominance features at the channel level.
3. A Color Distribution Alignment (CDA) module is proposed to enforce consistency in the color feature distribution, leading to clearer and more natural results.

**Weaknesses:**

1. The paper exhibits limited innovation, as its core innovative points bear striking similarities to those of CIDNet. In particular, the proposed VCF (Variance-aware Channel Filtering Stage) module closely resembles the LCA (Lighten Cross-Attention) module in CIDNet.
2. Comparative analysis of enhancement results for SID is lacking.
3. The ablation experiments are incomplete and thus fail to effectively evaluate the validity of the modules proposed in this paper.

**Questions:**

1. Why are the objective comparison results on the SID dataset provided, while the visual comparison of the enhanced results is not?

---

> ### Author Response · Authors · 2025-11-21
> **Response to Reviewer UKY5**
>
> Dear Reviewer UKY5:
>
> We sincerely appreciate your comments on our manuscript; they are extremely valuable to us.
> Below, we provide our detailed responses after carefully reflecting on the issues you raised. ：）
>
>
> **About Similarity**:
> Although both our method and CIDNet operate in the HVI space, the underlying motivations and points of innovation differ substantially. We approach low-light image enhancement from distinct perspectives, particularly regarding the LCA module in CIDNet and our VCF module. To provide an intuitive illustration, consider a classroom scenario in which a teacher asks all students to describe the content on the blackboard. The LCA module focuses on improving how accurately each student describes the blackboard, whereas our VCF module aims to select a subset of students who are inherently better at articulating the blackboard content. While both methods share the same high-level objective, they represent different forms of reasoning about the problem. We believe this demonstrates a deeper extension and multidimensional optimization of the task.
>
> In addition, to further demonstrate the differences between our method and CIDNet, we introduce the HSV space as an alternative to the HVI space and evaluate both approaches on the LOLv2-Real dataset. The results (see Figure 5(a)) are:
>
> | Methods      | PSNR↑  | SSIM↑ |
> | ------------ | ------ | ----- |
> | HVI (CIDNet) | 24.111 | 0.871 |
> | HVI (Ours)   | 24.758 | 0.893 |
> | HSV (CIDNet) | 21.349 | 0.801 |
> | HSV (Ours)   | 22.104 | 0.853 |
>
>
> We observe that our method improves image quality in both HVI and HSV spaces, which demonstrates its reliability and strong generalization capability.
>
> **About Visual**:
> We fully agree with your observation regarding the visualization issues on the SID dataset. In response, we have added the corresponding visual results in Figure 9(b).
>
>
> **About Ablation**:
> We have already included several ablation studies in the manuscript. In addition, we have further expanded the ablation experiments to provide a more comprehensive evaluation of the contribution of each individual module. Below are the additional ablation studies we conducted to more comprehensively evaluate the contribution of each component.
>
> Table 1: Metrics for Different Masking Ratios
> | Masking Ratio | PSNR (↑) | SSIM (↑) | LPIPS (↓) |
> | ------------- | -------- | -------- | --------- |
> | Ratio = 1/5   | 23.997   | 0.853    | 0.141     |
> | Ratio = 1/4   | 24.516   | 0.872    | 0.126     |
> | Ratio = 1/3   | 24.758   | 0.893    | 0.105     |
> | Ratio = 1/2   | 22.963   | 0.804    | 0.207     |
>
> Table 2: Number of CAA Modules
> | x     | PSNR (↑) | SSIM (↑) |
> | ----- | -------- | -------- |
> | x = 1 | 24.76    | 0.892    |
> | x = 2 | 24.74    | 0.891    |
> | x = 3 | 24.74    | 0.892    |
> | x = 4 | 24.74    | 0.892    |
> | x = 5 | 24.76    | 0.892    |
>
>
> Ablation of TCE on LOLv2-Real
> | Method           | PSNR (↑) | SSIM (↑) |
> | ---------------- | -------- | -------- |
> | w/o TCE          | 24.683   | 0.874    |
> | TCE-Vanilla Conv | 24.702   | 0.889    |
> | Ours             | 24.758   | 0.893    |
>
>
> Three-Branch Ablation
> | Method     | PSNR (↑) | SSIM (↑) |
> | ---------- | -------- | -------- |
> | w/o (F_t1) | 24.613   | 0.881    |
> | w/o (F_t2) | 24.562   | 0.879    |
> | w/o (F_t3) | 24.598   | 0.877    |
> | Ours       | 24.758   | 0.893    |
>
>
> Thank you for your careful review and valuable feedback on our work. Our method explores existing challenges and prior approaches from a different perspective, and we greatly appreciate the insightful comments you have provided. If you have any further questions or suggestions, we are more than willing to continue the discussion and offer additional information. We remain open to your feedback and are committed to further improving and refining our work.
>
> If you find our revisions satisfactory, we kindly ask you to consider adjusting your score accordingly based on your evaluation. We sincerely appreciate your time, effort, and support.

---

### Note · Authors · 2025-11-28

**Comment:**

We have carefully reviewed the reviewers’ comments and submitted our responses accordingly. However, we have not received any further updates or feedback. Therefore, we would like to withdraw our submission. Thank you for your time and consideration.

**Withdrawal Confirmation:**

I have read and agree with the venue's withdrawal policy on behalf of myself and my co-authors.